# FETV: A Benchmark for Fine-Grained Evaluation of Open-Domain Text-to-Video Generation

**Yuanxin Liu[§], Lei Li[§], Shuhuai Ren[§], Rundong Gao[¶], Shicheng Li[§],**
**Sishuo Chen[¶], Xu Sun[§], Lu Hou[‡]**

[§] National Key Laboratory for Multimedia Information Processing,
School of Computer Science, Peking University
[¶] Center for Data Science, Peking University    [‡] Huawei Noah's Ark Lab
{liuyuanxin, shuhuai_ren, gaord20}@stu.pku.edu.cn   nlp.lilei@gmail.com
{lisc99, chensishuo, xusun}@pku.edu.cn   houlu3@huawei.com

## Abstract

Recently, open-domain text-to-video (T2V) generation models have made remarkable progress. However, the promising results are mainly shown by the qualitative cases of generated videos, while the quantitative evaluation of T2V models still faces two critical problems. Firstly, existing studies lack fine-grained evaluation of T2V models on different categories of text prompts. Although some benchmarks have categorized the prompts, their categorization either only focuses on a single aspect or fails to consider the temporal information in video generation. Secondly, it is unclear whether the automatic evaluation metrics are consistent with human standards. To address these problems, we propose **FETV**, a benchmark for **F**ine-grained **E**valuation of **T**ext-to-**V**ideo generation. FETV is multi-aspect, categorizing the prompts based on three orthogonal aspects: the major content, the attributes to control and the prompt complexity. FETV is also temporal-aware, which introduces several temporal categories tailored for video generation. Based on FETV, we conduct comprehensive manual evaluations of four representative T2V models, revealing their pros and cons on different categories of prompts from different aspects. We also extend FETV as a testbed to evaluate the reliability of automatic T2V metrics. The multi-aspect categorization of FETV enables fine-grained analysis of the metrics' reliability in different scenarios. We find that existing automatic metrics (e.g., CLIPScore and FVD) correlate poorly with human evaluation. To address this problem, we explore several solutions to improve CLIPScore and FVD, and develop two automatic metrics that exhibit significant higher correlation with humans than existing metrics. Benchmark page: `https://github.com/llyx97/FETV`.

## 1 Introduction

Open-domain text-to-video (T2V) generation, which involves synthesizing videos in response to free-form textual prompts, has experienced significant advancements in recent years. Contemporary T2V models have demonstrated impressive visual quality and alignment with prompts. However, this progress is mainly supported by the qualitative cases of generated videos. To accurately assess the current state of T2V generation, enable model comparisons, and guide future research, it is essential to develop robust quantitative evaluation systems

However, existing quantitative evaluations for T2V models present two major challenges: **(1) Limited scope of performance measurements.** Current studies typically report overall results on entire test sets, providing only a coarse view of model performance. A more nuanced, fine-grained evaluation is necessary to gain a comprehensive understanding of model capabilities. While some benchmarks categorize text prompts based on content or challenge type [31, 50], they often neglect the temporal

Table 1: Comparison of text-to-image/video generation benchmarks. ✗ and ✓ denote whether the benchmark contains the corresponding aspect (column) of categorization or whether it is open-domain. FETV is open-domain, multi-aspect and temporal-aware, compared with other benchmarks.

| Task Type | Benchmark | Open Domain | Major Content | | Attribute Control | | Prompt Complexity |
| --- | --- | --- | --- | --- | --- | --- | --- |
| | | | spatial | temporal | spatial | temporal | |
| Text2Image | DrawBench | ✓ | ✗ | ✗ | ✓ | ✗ | ✗ |
| | PartiPrompts | ✓ | ✓ | ✗ | ✓ | ✗ | ✓ |
| Text2Video | UCF-101 | ✗ | ✗ | ✓ | ✗ | ✗ | ✗ |
| | Kinetics | ✗ | ✗ | ✓ | ✗ | ✗ | ✗ |
| | MSR-VTT | ✓ | ✓ | ✗ | ✗ | ✗ | ✗ |
| | Make-a-Video-Eval | ✓ | ✓ | ✗ | ✗ | ✗ | ✗ |
| | FETV (Ours) | ✓ | ✓ | ✓ | ✓ | ✓ | ✓ |

aspects of video generation. **(2) Lack of reliable automatic evaluation metrics.** In text-to-image generation, it has been found that the existing automatic evaluation metrics are inconsistent with human judgment [6, 27, 28]. However, in T2V generation, this problem is still under-explored.

To address these issues, we introduce a benchmark called **FETV** for the **F**ine-grained **E**valuation of **T**ext-to-**V**ideo generation, which offers a comprehensive categorization system for T2V evaluation. FETV includes a diverse set of text prompts, categorized based on three orthogonal aspects: major content, attribute control, and prompt complexity (see Figure 1). We also propose several temporal categories specifically designed for video generation, encompassing both spatial and temporal content. As shown in Table 1, compared with existing benchmarks, FETV highlights the most comprehensive categorization for fine-grained evaluation of T2V generation. To efficiently categorize the text prompts, we employ a two-step annotation process involving an automatic assignment of category labels followed by manual review. On this basis, we collect 619 categorized text prompts from existing open-domain text-video datasets [48, 1] and manually-written unusual prompts.

Using FETV, we perform comprehensive manual evaluations for four representative T2V models, enabling fine-grained analysis of their strengths and weaknesses across different prompt categories. The results highlight the weaknesses of existing T2V models in terms of (1) poor video quality for action and kinetic motion related videos and (2) inability to control quantity, motion direction and event order (see Section 3.1 for detailed definition of these categories). Furthermore, FETV can serve as a testbed for assessing the reliability of automatic evaluation metrics through correlation analysis with manual evaluations. Our experiments reveal that the widely used automatic metrics of video quality (FID [11] and FVD [40]) and video-text alignment (CLIPScore [10]) correlate poorly with human evaluation. In response to this problem, we explore several solutions to improve CLIPScore and FVD. Based on UMT [22], an advanced vision-language model (VLM), we develop FVD-UMT for video quality and UMTScore for video-text alignment, which exhibit much higher correlation with humans than existing metrics.

The contributions of this work are as follows: **(1)** We propose the FETV for fine-grained evaluation of open-domain T2V generation. Compared with existing benchmarks, FETV provides a more comprehensive view of T2V models' capabilities from multiple aspects. **(2)** Based on FETV, we conduct a comprehensive manual evaluation of four representative open T2V models and reveal their pros and cons from different aspects. **(3)** We extend FETV as a testbed to evaluate the reliability of automatic T2V metrics and develop two automatic metrics that exhibit significant higher correlation with humans than existing metrics.

## 2 Related Work

**Benchmarks for Text-to-Video Generation.** Most existing studies on T2V generation adopt datasets from other related tasks to evaluate their models. UCF-101 [38] and Kinetics-400/600/700 [16, 4, 36] are initially proposed for the task of human action recognition, which cannot cover the diversity of open-domain video generation. MSR-VTT [48], which is commonly used for video captioning and video-text retrieval, consists of text-video pairs covering 20 categories with diverse video contents. However, its categorization only considers the topic of videos (e.g., "music", "sports" and "news") while ignoring other important aspects in text-to-video generation, e.g., the temporal

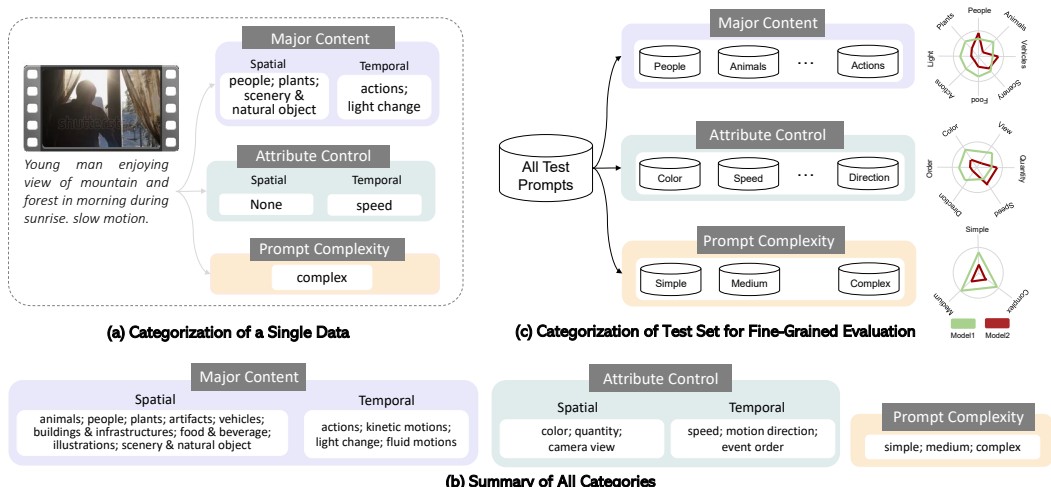

Figure 1: Illustration of our multi-aspect categorization (a-b), based on which we can realize fine-grained evaluation (c). Details of each category are shown in Figure 2 and Appendix B.1.

contents and the attributes a text prompt aims to control. In text-to-image generation, DrawBench [31] and PartiPrompts [50] introduce another aspect of categorization concerning the type of challenge a prompt involves. However, their prompts lack descriptions of temporal content, which are not suitable for T2V generation. Make-a-Video-Eval [34] contains 300 text prompts specially composed for T2V generation. However, their categorization (i.e., animals, fantasy, people, nature and scenes, food and beverage) only considers the spatial content.

**Automatic Metrics for Text-to-Video Generation.** There are two types of automatic open T2V metrics, which evaluate the video quality and video-text alignment, respectively. For video quality, the widely used metrics are the Inception Score (IS) [32], Frechet Inception Distance (FID) [11], and Frechet Video Distance (FVD) [40]. Based on a classification model, IS assesses whether every single generated video contains identifiable objects and whether all the videos are diverse in the generated objects. FID and FVD compute the distribution similarity between real and generated videos in the latent space of a deep neural network. In terms of video-text alignment, existing works on open T2V mainly adopt the CLIPScore [10]. CLIPScore is initially proposed to measure the similarity between machine-generated captions and the given image, using the pre-trained CLIP model [29]. In the task of text-to-image generation, it has been found that the automatic image quality and image-text alignment metrics are inconsistent with human judgment [6, 27, 28]. When it comes to open T2V, this problem remains under-explored.

**Models for Text-to-Video Generation.** Existing text-to-video generation models can be divided into the Transformer-based models [45, 46, 15, 43] and the Diffusion-based models [14, 13, 34, 51, 17, 25, 2, 9, 44]. The former first converts the videos into discrete visual tokens and then trains the Transformer model to generate these visual tokens. The latter generates videos using the Diffusion Models [37, 12, 49], either on the original pixel space or a latent space. In this paper, We evaluate three models that are open-sourced up to the time of writing this paper, namely CogVideo [15], Text2Video-zero [17] and ModelScopeT2V [44], as well as a more recent model ZeroScope [39]. Note that text-driven video editing [26, 47], where the input is a text prompt and a source video, may also be referred to as text-to-video generation. In this paper, we focus on the typical kind of text-to-video generation, where the input condition is only a piece of text prompt.

## 3 FETV Benchmark

FETV is a dataset of text prompts divided into different categories from multiple aspects for fine-grained evaluation of T2V generation. In this section, we will introduce the multi-aspect categorization, how to collect the prompts and category labels and the application scope of FETV.

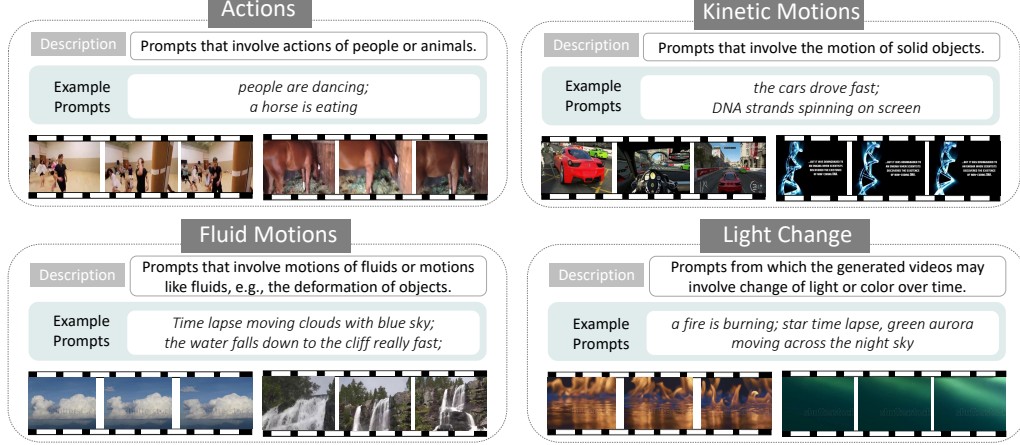

(a) Temporal categories under the "major content" aspect.

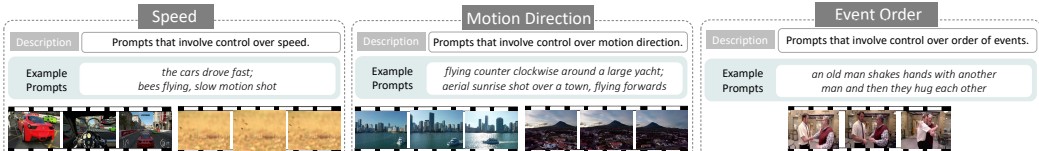

(b) Temporal categories under the "attribute control" aspect.

Figure 2: Descriptions and examples of the temporal categories.

## 3.1 Multi-Aspect Categorization

As shown in Figure 1, each prompt is categorized based on three aspects, namely the major content it describes, the attribute it aims to control in the generation of videos, and its complexity. The "major content" aspect is further divided into spatial and temporal categories, the former focus on the type of objects described in the prompt, while the latter focus on the type of temporal content. Similarly, the "attribute control" aspect is comprised of both spatial attributes and temporal attributes. The "prompt complexity" aspect has three levels of complexity, namely "simple", "medium", and "complex", depending on the number of non-stop words contained in a prompt. With the multi-aspect categorization of every single prompt, the entire FETV benchmark can be divided into different subsets (Figure 1 (c)), therefore enabling fine-grained evaluation. Figure 2 presents the descriptions and corresponding examples of the temporal categories. The complete description of all categories can be found in Appendix B.1.

## 3.2 Data Collection

**Prompt Sources.** The prompts of FETV come from two kinds of sources. **(1)** we reuse the texts from existing datasets of open-domain text-video pairs, which guarantees the diversity of prompts. Specifically, we use the MSR-VTT [48] test set and WebVid [1] validation set, which contain 59,800 and 4,999 text-video pairs, respectively. **(2)** we manually write prompts describing scenarios that are unusual in the real world, which cannot be found in existing text-video datasets. These "unusual prompts" are inspired by the "Imagination" category in PartiPrompts [50] and "Conflicting" category in DrawBench [31], which aims to test the generalization ability of the generation models. To write the unusual prompts, we first define several types of unusual cases, namely "object-action", "object-object", "object-color", "object-quantity", "object-direction", "object-speed" and "event order". For example, the prompt "A cat is driving a car." belongs to the case of unusual "object-action". Then, for each case, we enumerate several specific objects and attributes and write the prompts.

**Prompt Categorization and Selection** For the prompts from both sources, we assign category labels to them based on our multi-aspect categorization structure. This is achieved in two steps: **In the first step**, we define a list of keyphrases, WordNet synsets and some hand-crafted matching patterns for each category and automatically categorize the prompts. For example, to determine

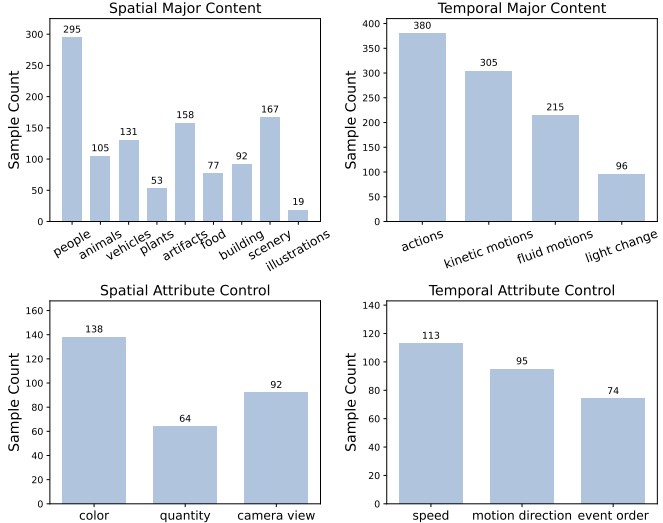

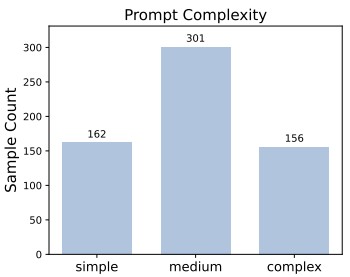

Figure 3: Data distribution over categories under the "major content" (upper) and "attribute control" (lower) aspects.

Figure 4: Data distribution over categories under the "prompt complexity" aspect.

whether a prompt belongs to the "animals" category, we iterate over the words in the prompt and see whether any of them belong to the animal-related WordNet synset. Details of the automatic matching rules for each category are presented in Appendix G. **In the second step**, we manually select prompts for each category and make some revisions to the category labels to ensure that they are correct. Specifically, we enumerate combinations of $(c_i, c_j)$, where $c_i$ belongs to the six "attribute control" categories and $c_j$ belongs to the four "temporal major content" categories. For each combination, we select roughly 20 prompts that belong to both $c_i$ and $c_j$, according to the first-step categorization, and then revise the incorrect category labels. For the text prompts from existing datasets, we also revise the ones that do not align well with the reference video.

## 3.3 Dataset Statistics and Data Format

We collect a total of 619 prompts with corresponding categorizations, among which 541 prompts are from existing datasets and 78 unusual prompts are written by us. Such data scale is comparable with existing text-to-image/video benchmarks, e.g., DrawBench (200 prompts), PartiPrompts (1,600 prompts) and Make-a-Video-Eval (300 prompts), which are mainly proposed for manual evaluation. The data distributions over categories under different aspects are illustrated in Figure 3 and Figure 4, which show that FETV contains enough data to cover the diverse range of categories. We also compare FETV with MSR-VTT and WebVid in terms of the data distribution over categories, which can be found in Appendix B.2.

Each data sample in FETV consists of three elements: the prompt $t$, the reference video $v^{\text{ref}}$ (not available for the unusual prompts) and the categorization labels $\{\mathcal{C}^a\}_{a \in \mathcal{A}}$. $\mathcal{A}$ is the collection of the three aspects, $\mathcal{C}^a$ is the categorization labels of $t$ under aspect $a$. Taking the prompt in Figure 1 as an example, when the aspect $a$ is "attribute control", $\mathcal{C}^a = \{\text{"speed"}\}$. Note that except for the complexity aspect, we allow each prompt to be classified into multiple categories under the same aspect. This setting is more realistic compared with existing benchmarks that only classify each prompt into a single category. More examples of FETV data can be found in Appendix B.3.

## 3.4 Application Scope

Theoretically, FETV can be used for both manual and automatic evaluation. However, considering that existing automatic T2V metrics correlate poorly with humans (see Section 5), we mainly rely on manual evaluation to obtain more accurate results. In addition to evaluating T2V models, FETV can also be extended to diagnose automatic metrics, using the manual evaluation as a reference.

# 4 Manual Evaluation of Open T2V Models

## 4.1 Models

We select four representative T2V models, which are open-sourced, for evaluation. We briefly describe them as follows and defer the details to Appendix C.

**CogVideo** [15] is based on the Transformer architecture [42]. It first transforms the video frames into discrete visual tokens using VQVAE [41] and then utilizes two Transformer models to hierarchically generate the visual tokens from a given text prompt.

**Text2Video-zero** [17] is based on the Diffusion Probabilistic Model [37]. It generates videos by enhancing the latent codes of a text-to-image Stable Diffusion model [1] [30] with motion dynamics, without training the model on any video data.

**ModelScopeT2V** [44] is another diffusion-based model. It extends the T2I Stable Diffusion model with temporal convolution and attention blocks and conducts training using both image-text and video-text datasets.

**ZeroScope** [39] is a watermark-free T2V generation model trained from the weights of ModelScopeT2V. It is optimized for 16:9 compositions.

## 4.2 Evaluation Setups

**Evaluation Criteria.**   We evaluate the generated videos mainly from four perspectives: **Static quality** focuses on the visual quality of single video frames. **Temporal quality** focuses on the temporal coherence of video frames. **Overall alignment** measures the overall alignment between a video and the given text prompt. **Fine-grained alignment**[2] measures the video-text alignment regarding specific attributes. For the first three perspectives, we score the videos based on a 1-5 Likert-scale judgement. For fine-grained alignment, we adopt 1-3 Likert-scale judgement. To facilitate the consistency between human evaluators, we carefully design the wording of questions and descriptions of each rating level. The agreement between our human evaluators is high, with Krippendorff's $\alpha$ [19] of 0.711, 0.770, 0.638 and 0.653 in the above four perspectives. In Appendix E, we summarize the details of our instruction document for manual evaluation and the annotation interface.

**Implementation Details.**   We generate three videos for each prompt using the three models respectively. For each generated video and each of the above four perspectives, we collect three rating scores from graduate students in our laboratory and report the average rating. For the generated videos, we obtain 619 samples $\times$ 3 models $\times$ 3 perspectives $\times$ 3 humans =16,713 ratings for the first three perspectives and 5,184 ratings of fine-grained alignment perspective (note that not every prompt involves an attribute for evaluating fine-grained alignment). We also evaluate the 541 reference videos, which produce 6,219 ratings for the four perspectives. In total, we collect 28,116 ratings in our manual evaluation. To obtain the performance under a specific category, we report the average rating of all videos whose prompts belong to this category.

## 4.3 Results and Analysis

In this section, we present and analyse the results from a quantitative perspective (some qualitative analyses are shown in Appendix F.5). Specifically, we are interested in the following questions: (1) how video quality varies when generating different major contents, (2) how video-text alignment varies when controlling different attributes or facing different prompt complexity. We also investigate (3) how well the models generalize to unusual prompts in Appendix F.4.

The results of video quality are presented in Figure 5. We can observe that: (1) In terms of the spatial categories, **the generated videos containing "people" and "animals" have poor quality, while the videos containing "plants" and "scenery & natural objects" are of good quality**. This is because videos of the former two categories involve many high-frequency details (e.g., human faces and fingers) which are more difficult to learn than the low-frequency information in the latter two categories (e.g., water and clouds). (2) When it comes to the temporal categories, **the videos about "actions" and "kinetic motions" are of worse quality**, since the corresponding temporal content is of higher frequency. Although there is a correlation between some spatial and temporal categories

---

[1]We use the dreamlike-photoreal-2.0 version, following the default setup of Text2Video-zero

[2]In this context, we reuse the word "fine-grained" to describe alignment between a single video-text pair, while in other contexts it refers to the evaluation of T2V models on specific categories of prompts.

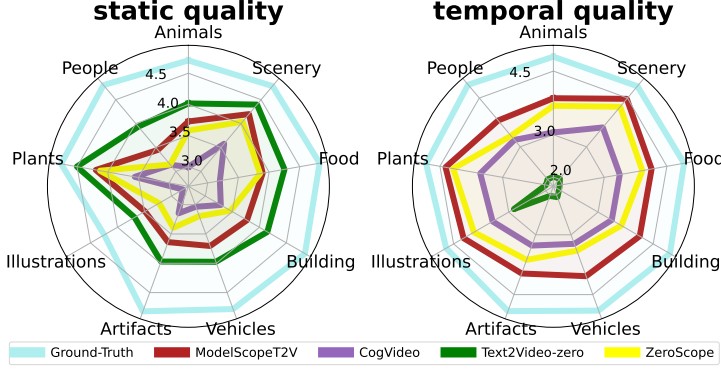

(a) Results across spatial major contents.

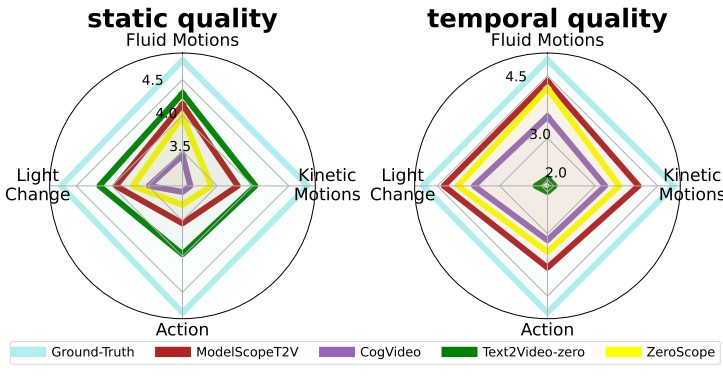

(b) Results across temporal major contents.

Figure 5: Manual evaluation of static and temporal video quality.

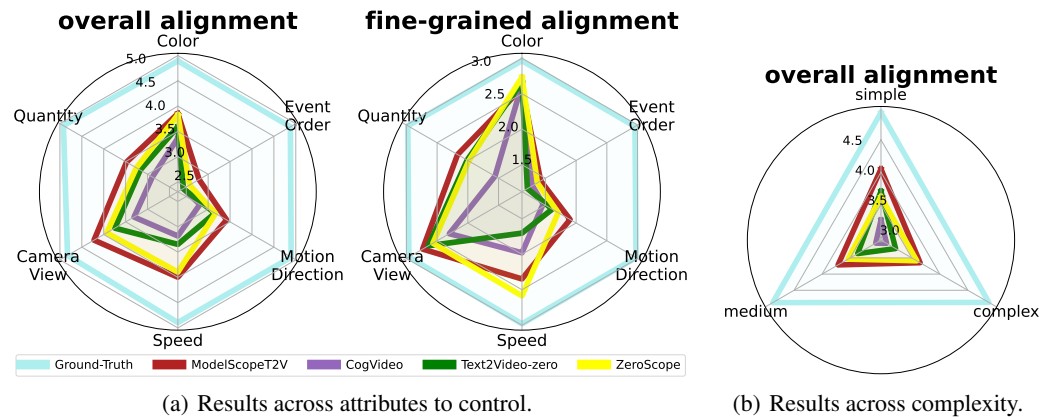

(a) Results across attributes to control.      (b) Results across complexity.

Figure 6: Manual evaluation of video-text alignment.

(e.g., "actions" is related to "people"), our temporal categorization provides a more direct view of the model performance from the temporal content perspective. (3) **None of the four models achieve comparable video quality with real videos**, especially in the aforementioned poor-quality categories. (4) Text2Video-zero performs the best in static quality while lagging far behind in temporal quality. This is because Text2Video-zero inherits the image generation power of Stable Diffusion, but the absence of video training prevents it from generating temporally coherent content. (5) ZeroScope, which is trained from ModelScopeT2V, is comparable with ModelScopeT2V in temporal quality

| Rank | Model | Score |
|------|-------|-------|
| 🥇 | Text2Video-Zero | 4.11 |
| 🥈 | ModelScopeT2V | 3.78 |
| 🥉 | ZeroScope | 3.52 |
| 4 | CogVideo | 3.24 |

**(a) Static Quality**

| Rank | Model | Score |
|------|-------|-------|
| 🥇 | ModelScopeT2V | 4.07 |
| 🥈 | ZeroScope | 3.72 |
| 🥉 | CogVideo | 3.30 |
| 4 | Text2Video-Zero | 2.04 |

**(b) Temporal Quality**

| Rank | Model | Score |
|------|-------|-------|
| 🥇 | ModelScopeT2V | 3.79 |
| 🥈 | ZeroScope | 3.57 |
| 🥉 | Text2Video-Zero | 3.42 |
| 4 | CogVideo | 3.10 |

**(c) Overall Alignment**

Figure 7: Leaderboard on FETV benchmark based on manual evaluation.

while performing worse in static quality. We conjecture that this is because the ZeroScope is primarily optimized for the 16:9 composition and watermark-free videos, instead of for better visual quality and video-text alignment.

The results of video-text alignment are shown in Figure 6. We can see that: (1) **Existing T2V models can already control "color" and "camera view" in most cases.** (2) **All four model struggle to accurately control "quantity", "motion direction" and "event order".** However, ModelScopeT2V exhibits some preliminary ability to control "speed" and "motion direction", which is not observed in CogVideo and Text2Video-zero. (3) ZeroScope slightly underperforms the original ModelScopeT2V in video-text alignment. We attribute this to the same underlying cause as explained for the video quality performance. (4) The relative performance among different attributes and T2V models is similar between fine-grained and overall alignment, with a few exceptions. This is because the overall alignment may inevitably be affected by other attributes. Therefore, **the fine-grained alignment is more accurate when evaluating controllability over specific attributes, but the overall alignment is an acceptable approximation**. (5) While video-text alignment is slightly lower for the "complex" prompts, **we cannot observe a clear correlation between prompt complexity and video-text alignment.**

In addition to the fine-grained evaluating results, we also summarize the leaderboard on the entire FETV benchmark in Figure 7.

## 5 Diagnosis of Automatic Evaluation Metrics

In this section, we perform automatic evaluations on FETV and diagnose their correlation with manual evaluations.

### 5.1 Metrics

For video-text alignment, we investigate five metrics. **(1) CLIPScore** [10] is widely used to evaluate T2V/I generation models. CLIPScore is based on CLIP [29], a vision-language model (VLM) pre-trained on large-scale text-image pairs. **(2) CLIPScore-ft** is based on a CLIP model fine-tuned on the video-text retrieval task using MSR-VTT data. **(3) BLIPScore** and **(4) UMTScore** replace CLIP with more advanced VLMs, i.e., BLIP [21] and UMT [22]. BLIP adopts bootstrapping language-image pre-training. UMT is pre-trained large-scale video-text data and we adopt the fine-tuned version on MSR-VTT. Then, we compute BLIPScore in the same way as CLIPScore. For UMT, we utilize the cross-modal decoder's video-text matching result and refer to this metric as UMTScore. We also adopt a Video Large Language Models (LLMs), i.e., Otter [20], to evaluate video-text alignment by formulating it as a Video QA task. Specifically, we first extract the key elements from the text prompt and generate corresponding yes-no questions via the Vicuna model [5]. Then, we feed the generated (or ground-truth) videos and questions into the Video LLM. The average number of questions that receive a positive answer is defined as the alignment score for a specific video-text pair. We name the Video LLM-based metric as **(5) Otter-VQA**.

For video quality metrics, we investigate **FID** [11] and **FVD** [40]. The original FVD adopts the I3D model [3] as a video encoder. We introduce FVD-UMT, an enhanced version of FVD that leverages the UMT vision encoder for video feature extraction. More details of these metrics and our implementations can be found in Appendix D.

Table 2: Correlation between automatic and manual evaluations of video-text alignment, measured by Kendall's $\tau_c$ (left) and Spearman's $\rho$ (right) coefficients. We also report the average inter-human correlation in the final row as a reference.

| | Color | Quantity | Camera View | Speed | Motion Direction | Event Order | All |
|---|---|---|---|---|---|---|---|
| CLIPScore | 0.157/0.218 | 0.147/0.202 | 0.254/0.345 | 0.178/0.246 | 0.141/0.194 | 0.177/0.248 | 0.177/0.243 |
| CLIPScore-ft | 0.206/0.287 | 0.250/0.340 | 0.293/0.402 | 0.203/0.280 | 0.254/0.348 | 0.157/0.221 | 0.224/0.309 |
| BLIPScore | 0.222/0.307 | 0.207/0.282 | 0.285/0.394 | 0.195/0.266 | 0.223/0.305 | 0.180/0.250 | 0.235/0.322 |
| Otter-VQA | 0.049/0.070 | 0.134/0.188 | 0.027/0.038 | 0.051/0.073 | 0.119/0.166 | 0.146/0.206 | 0.081/0.114 |
| UMTScore | **0.304/0.420** | **0.394/0.528** | **0.300/0.415** | **0.296/0.407** | **0.356/0.476** | **0.295/0.406** | **0.309/0.425** |
| Human | 0.547/0.702 | 0.647/0.784 | 0.447/0.595 | 0.539/0.683 | 0.619/.0.747 | 0.517/0.680 | 0.576/0.719 |

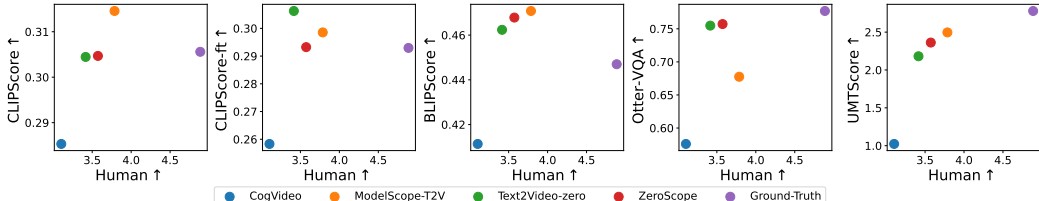

Figure 8: Automatic and human ranking of the T2V models in terms of video-text alignment. Results on specific categories are reported in Appendix F.1.

## 5.2 Evaluation Setups

**Video-Text Alignment Metrics.** We use the same generated and reference videos for manual evaluation (one video for a text prompt using one model) and compute the video-text alignment for each video-text pair. Then, we measure the correlation between automatic and manual evaluations, using Kendall's $\tau_c$ coefficient and Spearman's $\rho$ coefficient. To obtain the correlation under a specific category, we collect the manual and automatic evaluation results of video-text pairs that belong to this category and compute the coefficients.

**Video Quality Metrics.** Different from the video-text alignment metrics, FID and FVD evaluate the quality of a group of generated videos instead of a single video. To evaluate the fine-grained video quality in specific categories, for each open T2V model, we generate 300 videos for each category under the "major content" aspect, based on FETV prompts. In this case, each prompt can be used multiple times but the generated videos are different due to the randomness in the diffusion process and visual token sampling. We also sample 300 reference videos for each category. This is achieved by using the automatic categorization results of MSR-VTT and WebVid prompts (introduced in Section 3.2) and collecting the corresponding videos. To compute the overall FID and FVD for a T2V model, we sample 1,024 generated and reference videos (the impact of video sample number is analyzed in Appendix F.3). The results are averaged across 4 runs with different random seeds.

## 5.3 Results and Analysis

**Video-Text Alignment Metrics.** Table 2 shows the correlation between automatic and manual evaluations. We can see that: (1) Although the widely-used CLIPScore is positively correlated with manual evaluations, the overall correlation degree is very weak compared with inter-human correlations. (2) Fine-tuning CLIP on video-text retrieval and replacing CLIP with BLIP are beneficial, which promote the correlation across almost all categories. However, CLIPScore-ft and BLIPScore are still inconsistent with the human ranking of videos generated from different models, as shown in Figure 8. Therefore, we cannot rely on CLIPScore and BLIPScore to accurately evaluate open T2V models. (3) UMTScore exhibits the strongest correlation with humans when measured by Kendall's $\tau_c$ and Spearman's $\rho$. More importantly, it consistently aligns with human judgments in the T2V model rankings. The performance of UMTScore suggests that training VLMs with video data is important and the evaluation of open T2V models can benefit from the development of stronger VLMs. (4) The correlation varies across different "attribute control" categories. It is interesting to see that for UMTScore, the correlation in the challenging categories (for open T2V models), i.e., "Quantity" and "Motions Direction", is higher than in the relatively simple "Color" and "Camera

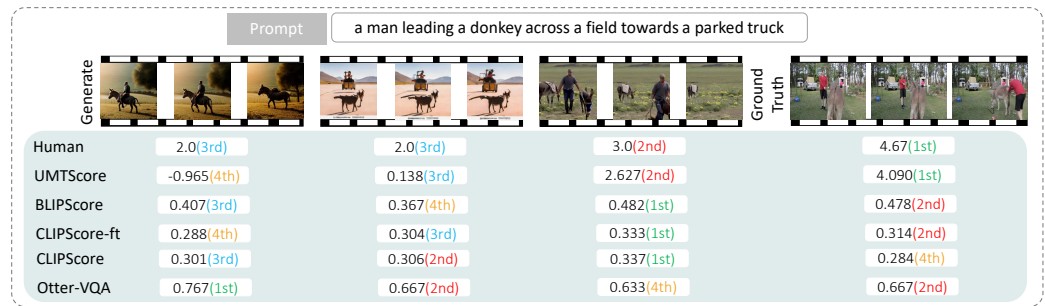

Figure 9: Video examples and alignment scores measured by different metrics. The left three videos are generated and the rightmost video is the ground-truth. More examples are shown in Appendix F.6.

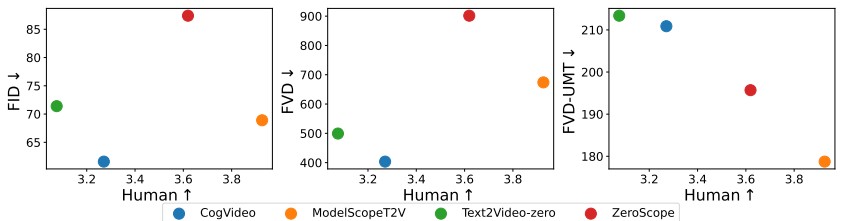

Figure 10: Automatic and human ranking of T2V models in terms of video quality (averaged static and temporal quality). Results on specific categories are reported in Appendix F.2.

View". We conjecture that this is because, in the simpler categories, the performance of open T2V models is more similar and it is more difficult to distinguish the difference, and vice versa.

Figure 9 presents a case study of video-text alignment scores. We can see that the results measured by CLIPScore and Otter-VQA strongly differ from human evaluation. CLIPScore-ft and BLIPScore correlate better with humans, while still mistakenly assigning lower score to the ground-truth video. In comparison, UMTScore's ranking of video-text alignment is most consistent with humans.

**Video Quality Metrics.** For FID and FVD, we cannot calculate the correlation coefficients because they are not sample-wise metrics. To investigate their correlation with humans, we focus on the system-wise ranking of open T2V models. As we can see in Figure 10, the FID and the original FVD with I3D video encoder are loosely coupled with the human's perception of the video quality. This result reveals the critical problem of using FID and FVD as a measurement of video quality, which is common in previous works. In comparison, the FVD-UMT metric, which is enhanced with the UMT model as video feature extractor, generates consistent rankings with humans.

## 6 Conclusions

In this paper, we propose the FETV benchmark for fine-grained evaluation of open-domain text-to-video generation. Compared with existing benchmarks, FETV is both multi-aspect and temporal-aware, which can provide more specific information on the performance of T2V models. Based on FETV, we conduct a comprehensive manual evaluation of representative T2V models and reveal their pros and cons in different categories of text prompts from different aspects. Furthermore, we extend FETV to diagnose the reliability of automatic T2V metrics, revealing the poor correlation of existing metrics with humans. To address this problem, we develop two automatic metrics using an advanced VLM called UMT, which can produce a consistent ranking of T2V models with humans.

## 7 Limitations and Future Work

The limitations of this work can be viewed from two perspectives. First, while the proposed UMT-based metrics are more reliable than existing automatic metrics, they still have room for improvement to better align with humans. Second, due to the lack of open-sourced open T2V models, the number of models evaluated in this work is limited. We invite follow-up studies to evaluate their models on our benchmark and encourage more projects to open-source their models.

## Acknowledgments and Disclosure of Funding

We thank all the anonymous reviewers for their constructive comments. This work is supported in part by a Huawei Research Grant and National Natural Science Foundation of China (No. 62176002). Xu Sun is the corresponding author of this paper.

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

# A  Overview of Appendix

# B  More Details of FETV

## B.1  Category Descriptions

In Figure 2 of the main paper, we illustrate the descriptions and examples of the temporal categories. In this appendix, Figure 11 presents the descriptions and examples of the spatial categories under the "major content" and "attribute control" aspects. Table 3 presents the descriptions and examples of different prompt complexity levels.

## B.2  Data Distribution

Figure 12 compares FETV with MSR-VTT and WebVid in terms of their data distribution over categories. To obtain the distribution of MSR-VTT and WebVid, we randomly sample 100 data from each dataset and manually annotate the categories as we construct FETV. As we can see, FETV exhibits a more uniform distribution over the categories, while MSR-VTT and WebVid are more biased toward certain categories. Particularly, more than 60% of the MSR-VTT and WebVid prompts do not involve the six attributes introduced in FETV. Consequently, the ability to control these categories, especially the challenging ones, cannot be reflected by the performance on MSR-VTT and WebVid (note that these two datasets lack categorization of the attributes).

## B.3  Data Examples

Table 4 shows FETV prompt examples with corresponding categorizations, which cover all 22 categories under the three aspects.

# C  Text-to-Video Generation Model Details

## C.1  CogVideo

**Model Architecture.**   CogVideo [15] generates videos as discrete visual tokens, encoded by the VQVAE[41], using the Transformer model [42]. The generation process consists of two stages. In the first stage, a Transformer sequentially generates some keyframes based on the text prompts. In the second stage, another Transformer generates the remaining frames by interpolating the generated frames in a recursive way. In both stages, CogVideo can generate videos according to a specified frame rate. CogVideo is initialized from the pre-trained text-to-image model CogView2 [6] and inserts a temporal channel attention to model the dynamics in videos. In total, the Transformer of both stages has 9.4 billion parameters, among which 6 billion parameters are from CogView2.

**Training Setups.**   CogVideo is trained on a private dataset of 5.4 million captioned videos with $160{\times}160$ spatial resolution. The first-stage Transformer is trained with a 1 fps frame rate and the second-stage Transformer is trained with 2,4,8 fps frame rates.

**Implementation Details.**   For inference of CogVideo, we adopt the official implementation with default hyper-parameters setups. We use the Transformer models of both stages and generate videos with 33 frames (a duration of 4 seconds), with a spatial resolution of $480{\times}480$ (upsampled by CogView2). The inference of CogVideo consumes approximately 24 GB of GPU memory. We use two types of GPU, i.e., Nvidia A40 and TITAN RTX. The inference of all four T2V models is run on a single GPU. The codes are based on the Pytorch framework.

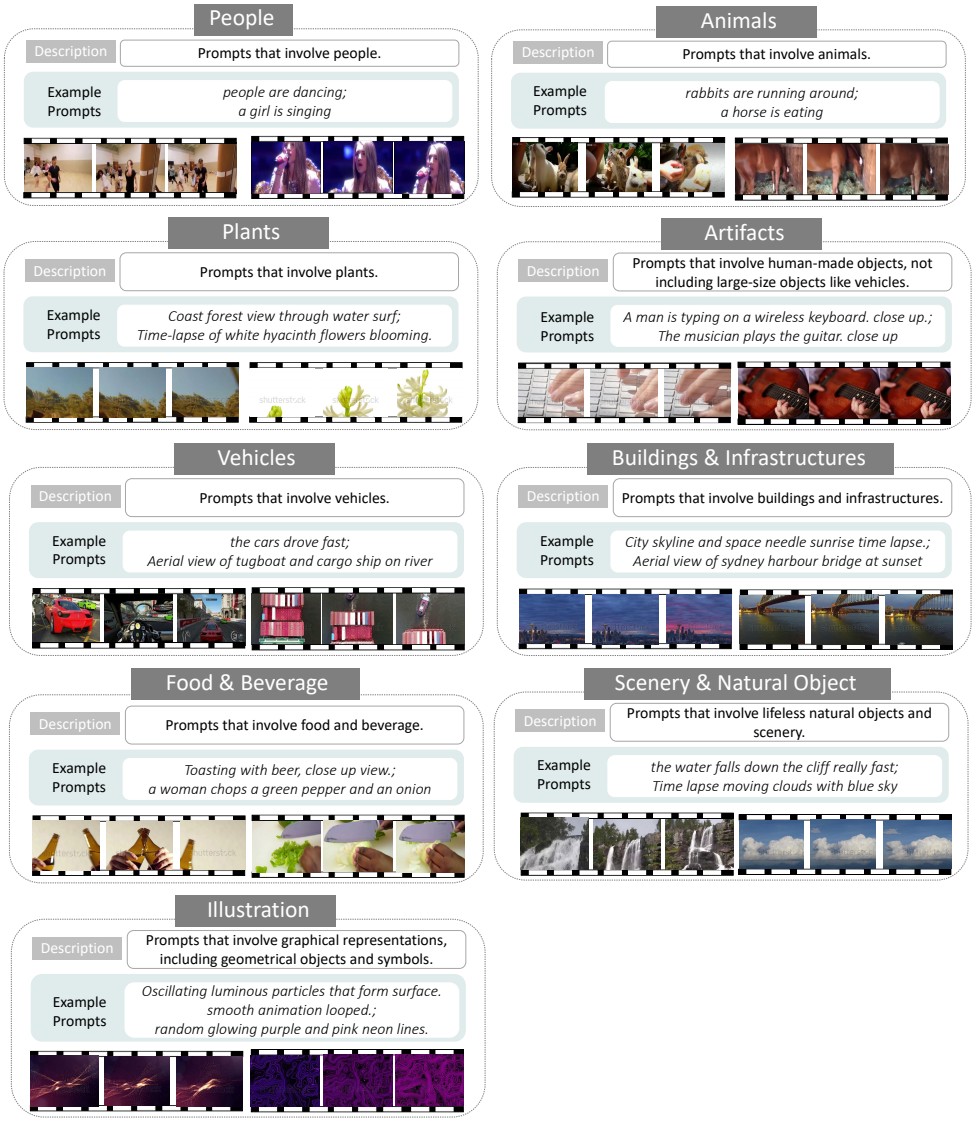

(a) Spatial categories under the "major content" aspect.

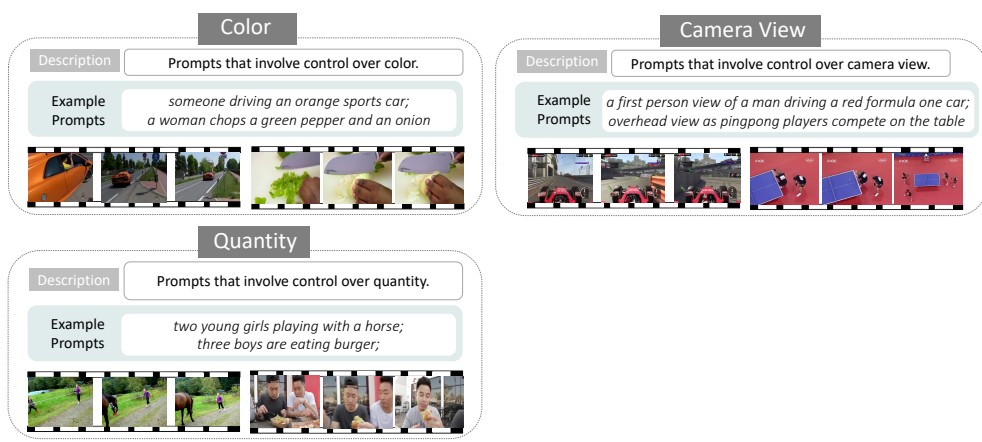

(b) Spatial categories under the "attribute control" aspect.

Figure 11: Descriptions and examples of the spatial categories.

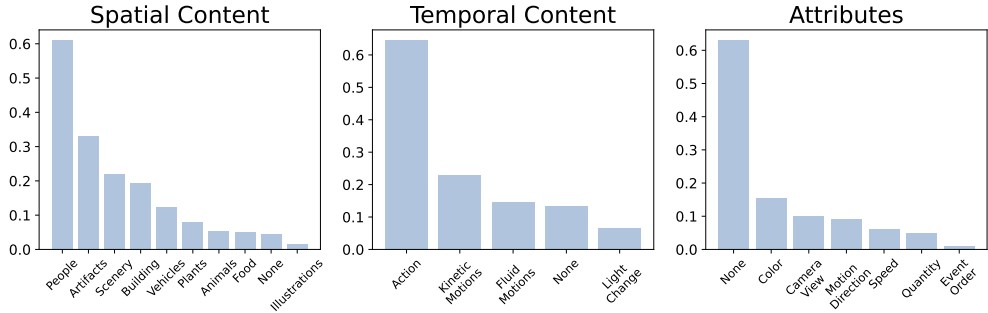

(a) Data distribution of 200 random samples from MSR-VTT and WebVid.

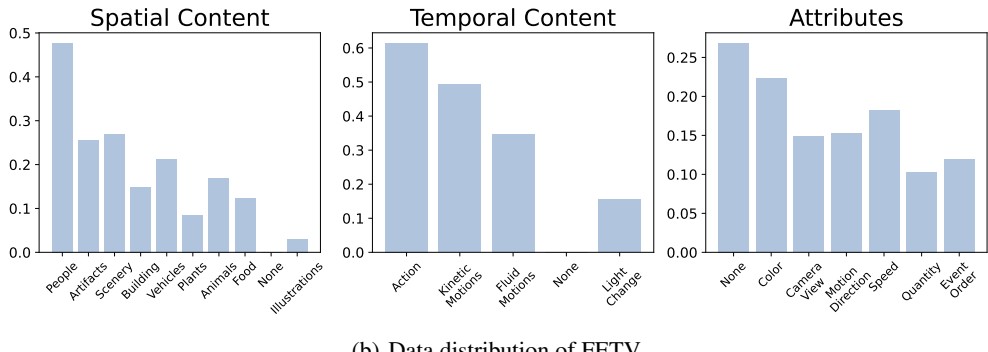

(b) Data distribution of FETV.

Figure 12: Percentile distribution of data across categories. "None" denotes the data samples that do not belong to any of the categories under certain aspects. Note that the distributions do not sum to one because each data sample can belong to multiple categories.

Table 3: Descriptions and examples of different "prompt complexity" categories.

| Category | Description | Example Prompts |
|---|---|---|
| **Simple** | Prompts that involve $0 \sim 4$ non-stop words. | Prompt1: *the cars drove fast*
Prompt2: *people are dancing*
Prompt3: *a dog is sneezing* |
| **Medium** | Prompts that involve $5 \sim 8$ non-stop words. | Prompt1: *A man is typing on a wireless keyboard . close up .*
Prompt2: *flying counter clockwise around a large yacht*
Prompt3: *aerial sunrise shot over a town, flying forwards* |
| **Complex** | Prompts that involve more than 8 non-stop words. | Prompt1: *Tilt down slow motion wide tracking shot*
*of man walking along stony ground towards clouds*
Prompt2: *in a cookery program the preparation of vegetable soup*
*is shown cutting leaves and add pinch of salt to water and boiling*
Prompt3: *a corn field moves quickly past the camera there is a*
*shot of a city and then of a double-decker passenger train* |

## C.2 Text2Video-Zero

**Model Architecture.** Text2Video-Zero [17] adapts text-to-image generation models for zero-shot text-to-video generation without any training on video data. Specifically, it introduces three techniques: (1) injecting motion dynamics into the first frame's latent code to obtain a sequence of dynamic frames; (2) replacing each frame's self-attention with cross-attention to the first frame to enhance temporal coherence; (3) a background smoothing strategy to enhance the consistency of background across frames.

**Implementation Details.** Text2Video-Zero is compatible with a wide range of diffusion-based text-to-image models. In this work, we utilize the dreamlike-photoreal-2.0 version of Stable Diffusion, following the default setup of Text2Video-Zero's official implementation. We generate videos of 2

Table 4: Prompts and categorizations of FETV data examples. "Fluid" denotes fluid motions; "Kinetic" denotes kinetic motions; "Light" denotes light change; "Buildings" denotes buildings & infrastructures; "Food" denotes food & beverage; "Scenery" denotes scenery & natural objects.

| Prompt | Major Content | | Attribute Control | | Prompt Complexity |
|---|---|---|---|---|---|
| | Spatial | Temporal | Spatial | Temporal | |
| *a person is cooking* | People | Actions | None | None | Simple |
| *rabbits are running around* | Animals | Actions | None | None | Simple |
| *someone folds paper* | Artifacts | Actions | None | None | Simple |
| *A mountain stream* | Scenery | Fluid | None | None | Simple |
| *a fire is burning* | Scenery | Light | None | None | Simple |
| *sea side road* | Scenery;Buildings | Fluid | None | None | Simple |
| *someone driving an orange sports car* | People;Vehicles | Actions;Kinetic | Color | None | Medium |
| *girl in pink dress fashion model walking in ramp* | People;Artifacts | Actions;Kinetic | Color | None | Medium |
| *Rows of young green sunflower plants in field in sunset, agriculture in spring* | Plants | Light | Color | None | Complex |
| *three boys are eating burger* | People;Food | Actions | Quantity | None | Simple |
| *four airplanes flying* | People;Vehicles | Actions;Kinetic | Quantity | None | Simple |
| *a first person view of a man driving a red formula one car* | People;Vehicles | Actions;Kinetic | Color;Camera | None | Complex |
| *overhead view as pingpong players compete on the table* | People;Artifacts | Actions | Camera | None | Medium |
| *Toasting with beer, close up view.* | Food | Fluid | Camera | None | Simple |
| *Bees flying, slow motion shot* | Animals | Actions;Kinetic | None | Speed | Medium |
| *the cars drove fast* | Vehicles | Kinetic | None | Speed | Simple |
| *Time lapse moving clouds with blue sky* | Scenery | Fluid | Color | Speed | Medium |
| *Flying counter clockwise around a large yacht* | Vehicles | Kinetic | None | Direction | Medium |
| *A wave approaching and crashing on to a big rock on the shore.* | Scenery | Fluid | None | Direction | Medium |
| *a man walks away from an suv* | People;Vehicles | Actions;Kinetic | None | Direction | Simple |
| *Abstract circular light pattern, artistic blurred background, camera moving from left to right* | Illustrations | Kinetic;Light | None | Direction | Complex |
| *an old man shakes hands with another man and then they hug each other* | People | Actions | None | Order | Medium |
| *a guy places a pan into tin foil and then places the pan into a water bath* | People;Artifacts | Actions;Fluid | None | Order | Complex |
| *a car first drives on a mountain road and then on a highway along the lake* | Vehicles;Scenery; Buildings | Kinetic | None | Order | Medium |

seconds at 8 fps, with a spatial resolution of 512×512. The inference of Text2Video-Zero consumes approximately 24 GB of GPU memory.

## C.3 ModelScopeT2V

**Model Architecture.**    ModelScopeT2V [44] extends the T2I Stable Diffusion model with temporal convolution and attention blocks, which enables video modelling. Then, the model is trained using both video-text and image-text datasets, which mitigates catastrophic forgetting of the T2I generation ability.

**Training Setups.** ModelScopeT2V is trained on both video-text pairs (WebVid-10M [1]) and image-text pairs (LAION2B-en [33]). During training, 1/8 of GPUs are allocated to images and the remaining 7/8 GPUs are used for videos.

**Implementation Details.** We implement ModelScopeT2V using the code and model checkpoints released in ModelScope. We generate videos of 2 seconds at 8fps, with a spatial resolution of $256\times256$. The inference of ModelScopeT2V consumes approximately 16 GB of GPU memory.

### C.4   ZeroScope

**Model Architecture and Training Setups.** ZeroScope [39] is a watermark-free video generation model optimized for producing 16:9 compositions. It is trained from the weights of ModelScopeT2V using 9,923 clips and 29,769 tagged frames at 24 frames, 576x320 resolution.

**Implementation Details.** We adopt the ZeroScope-v2-576w model, following the official implementation in HuggingFace. The generated videos are 3 seconds at 8fps, with a spatial resolution of 576x320. The inference of ZeroScope-v2-576w consumes approximately 20 GB of GPU memory.

## D   More Details of Automatic Evaluation Setups

### D.1   Video-Text Alignment Metrics

For the basic version of CLIPScore, we adopt the implementation from [18], with ViT-B/32 [7] as the visual encoder backbone. For BLIPScore, we utilize the BLIP model[3] [21], with ViT-B/16 as the backbone, fine-tuned on the COCO dataset [23] for image-text retrieval. For CLIPScore-ft, we fine-tune the ViT-B/32 CLIP model for video-text retrieval on the 7k training split of MSR-VTT [8], which has no overlap with the reference videos in FETV. The implementation and hyper-parameters settings follow CLIP4CLIP [24]. For these three automatic metrics, we average the image embeddings over 16 uniformly sampled frames to compute the similarity between video and text embeddings.

UMT [22] is composed of a video encoder, a text encoder and a cross-modal decoder. We adopt the large version of UTM pre-trained on 25M video-text pairs using four types of losses: Unmasked Token Alignment (UTA), Video-Text Contrastive (VTC) learning, Video-Text Matching (VTM) and Masked Language Modeling (MLM), with 4 video frames as input. Then it is fine-tuned on MSR-VTT using VTC and VTM, with 12 video frames as input. We utilize the cross-modal decoder's VTM result to compute the UTMScore.

To extract elements from video captions and generate questions for Video QA, we utilize the 13B version of Vicuna v1.3. Table 5 shows some examples of extracted elements and generated questions. The full list of elements and questions for the entire FETV dataset and prompts to Vicuna can be found in this link. For Otter-VQA, we employ the video version of Otter, which is based on LLaMA7B. We deem the Video LLM's response as positive if it begins with "yes" and negative otherwise. Then, the average number of questions that receive a positive answer is defined as the alignment score for a specific video-text pair.

### D.2   Video Quality Metrics

As mentioned in Section 4.2, we collect 300 generated and reference videos of a particular category to compute FID [11] and FVD[4] [40]. Specifically, to collect the generated videos, we randomly sample from the 619 prompts in FETV (a prompt can be sampled multiple times), until every major content category has at least 300 sampled prompts, and generate videos using the T2V models. To collect the reference videos, we randomly sample from MSR-VTT test set (59,800 prompts) and WebVid validation set (4,999 prompts), until every major content category has at least 300 sampled prompts, and collect the corresponding videos. In this way, we collect 2,055 generated videos (for each T2V model) and 2,328 reference videos. Figure 13 summarizes the distribution of the sampled prompts. To compute FID and FVD for a particular category, we sample 300 generated and reference videos that belong to this category, and average over 4 runs with different random seeds.

---

[3]`https://github.com/salesforce/BLIP`
[4]We adopt clean-fid's [28] implementation of FID and stylegan-v's [35] implementation of FVD.

Table 5: Examples of extracted elements and generated questions by Vicuna.

**Video Description**: people are dancing
**Extracted Elements**: people, dancing
**Generated Questions**:
- About people: are there people in the video?
- About dancing: are the people in the video dancing?

**Video Description**: Time lapse moving clouds with blue sky
**Extracted Elements**: time lapse, clouds, moving, blue sky
**Generated Questions**:
- About time lapse: is this a time lapse?
- About clouds: are there clouds in the video?
- About moving: are the clouds moving in the video?
- About blue sky: is the video showing a blue sky?

**Video Description**: cartoon characters driving a plane across a rainbow to the right
**Extracted Elements**: cartoon characters, driving, plane, rainbow, right
**Generated Questions**:
- About cartoon characters: are there cartoon characters?
- About driving: are the cartoon characters driving?
- About plane: is this video about a plane?
- About rainbow: is the video showing a rainbow?
- About right: is the plane driving to the right?

Table 6: Inter-human correlation in manual evaluation. Kendall's $\tau_c$ and Spearman's $\rho$ are averaged across the correlation between every two humans. $\pm$ denotes standard deviations. The p-values are less than 0.05.

| | Static Quality | Temporal Quality | Overall Alignment | Fine-grained Alignment |
|---|---|---|---|---|
| Kendall's $\tau_c$ | 0.55±0.04 | 0.62±0.06 | 0.58±0.03 | 0.58±0.03 |
| Spearman's $\rho$ | 0.70±0.05 | 0.74±0.06 | 0.72±0.04 | 0.72±0.04 |
| Krippendorff's $\alpha$ | 0.689 | 0.712 | 0.674 | 0.730 |

## E    More Details of Manual Evaluation Setups

### E.1    Evaluation Instructions

[27] has shown that a specific definition of each rating level can facilitate the inter-human correlation in the manual evaluation of T2I models. Therefore, we carefully design the rating level definitions and provide some examples of ratings as a reference. Figure 14 and Figure 15 show the instructions for manual evaluation of video quality and video-text alignment, respectively.

### E.2    Evaluation Interface

Figure 16 illustrates our manual evaluation interface, which is based on the Python tkinter package. To alleviate the influence of resolution in the assessment of video quality, we crop all the generated and reference videos to 256×256 spatial resolution in the interface. Typically, the reference videos are longer than the generated ones. To prevent the reference videos from being identified based on the duration, we crop them to 4 seconds when evaluating video quality. When evaluating video-text alignment, the reference videos are kept to the original duration to maintain all the content described in the text prompts. Considering that the quality of recently rated videos may influence the assessment of the current video (e.g., if the recently rated videos are of poor quality, the evaluator may be inclined to lower the standard), we alternately display videos generated by different models.

### E.3    Correlation between Human Evaluators

The correlation between human evaluators is shown in Table 6, which demonstrates strong inter-human agreement across the four perspectives to evaluate.

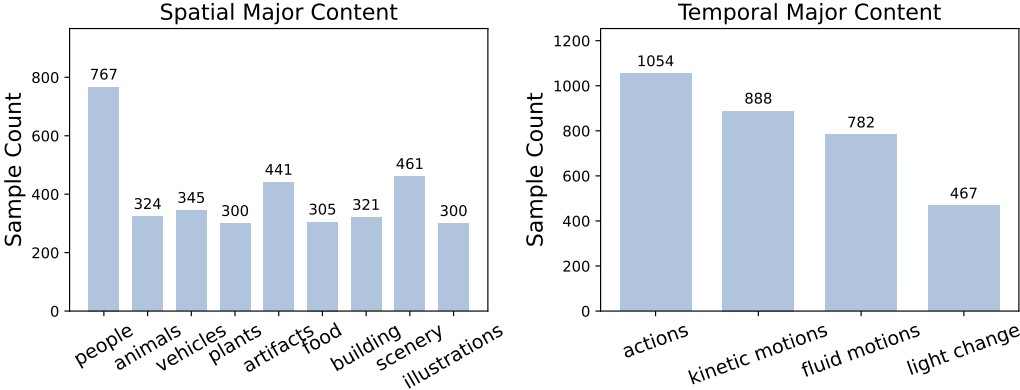

(a) Distribution of data sampled from FETV to generate videos.

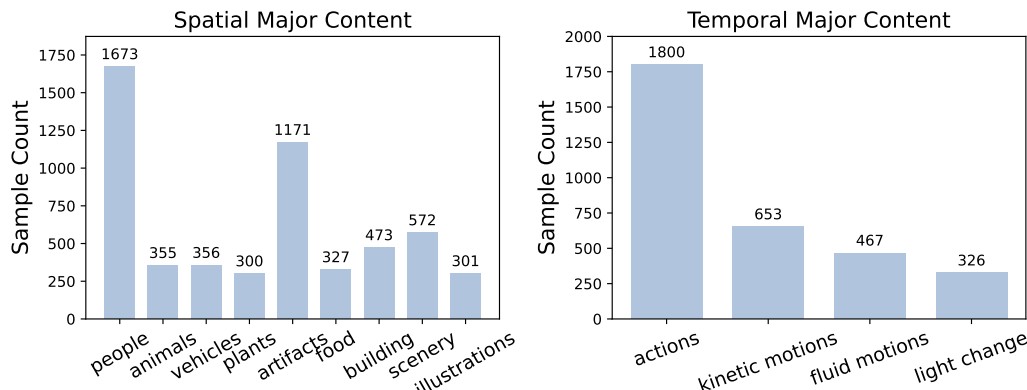

(b) Distribution of data sampled from MSR-VTT and WebVid to collect reference videos.

Figure 13: Sampled data distribution for computing FID and FVD.

# F  More Experimental Results

## F.1  Video-Text Alignment Ranking Results

In Figure 8 of the main paper, we compare automatic and human ranking of the T2V model in terms of video-text alignment over the entire FETV benchmark. The results in specific attribute control categories are shown in Figure 17. It can be seen that UMTScore exhibits the strongest correlation with humans in every category, as compared with other automatic alignment metrics.

## F.2  FID and FVD Ranking Results

In Figure 10 of the main paper, we compare FID/FVD and human ranking of the T2V model in terms of video quality over the entire FETV benchmark. Figure 18 and Figure 19 present the results in specific spatial and temporal categories. We can find that, compared with FID and FVD, FVD-UMT is more consistent with human rankings in almost all categories. However, in some categories (e.g., "Illustration", "Food & Beverage" and "Plants") all three automatic metrics are poorly aligned with humans. For these categories, we still need to rely on manual evaluation as the golden standard of video quality.

## F.3  The Effect of Video Number on FID and FVD

As shown in Figure 20, the number of videos has a substantial impact on the values of FID and FVD, which decrease with the increase of video number. However, when the number of videos exceeds 300, FVD-UMT can provide more accurate ranking of the T2V models, as compared with FID and FVD. Since generating thousands of videos is time-consuming (especially for CogVideo which takes

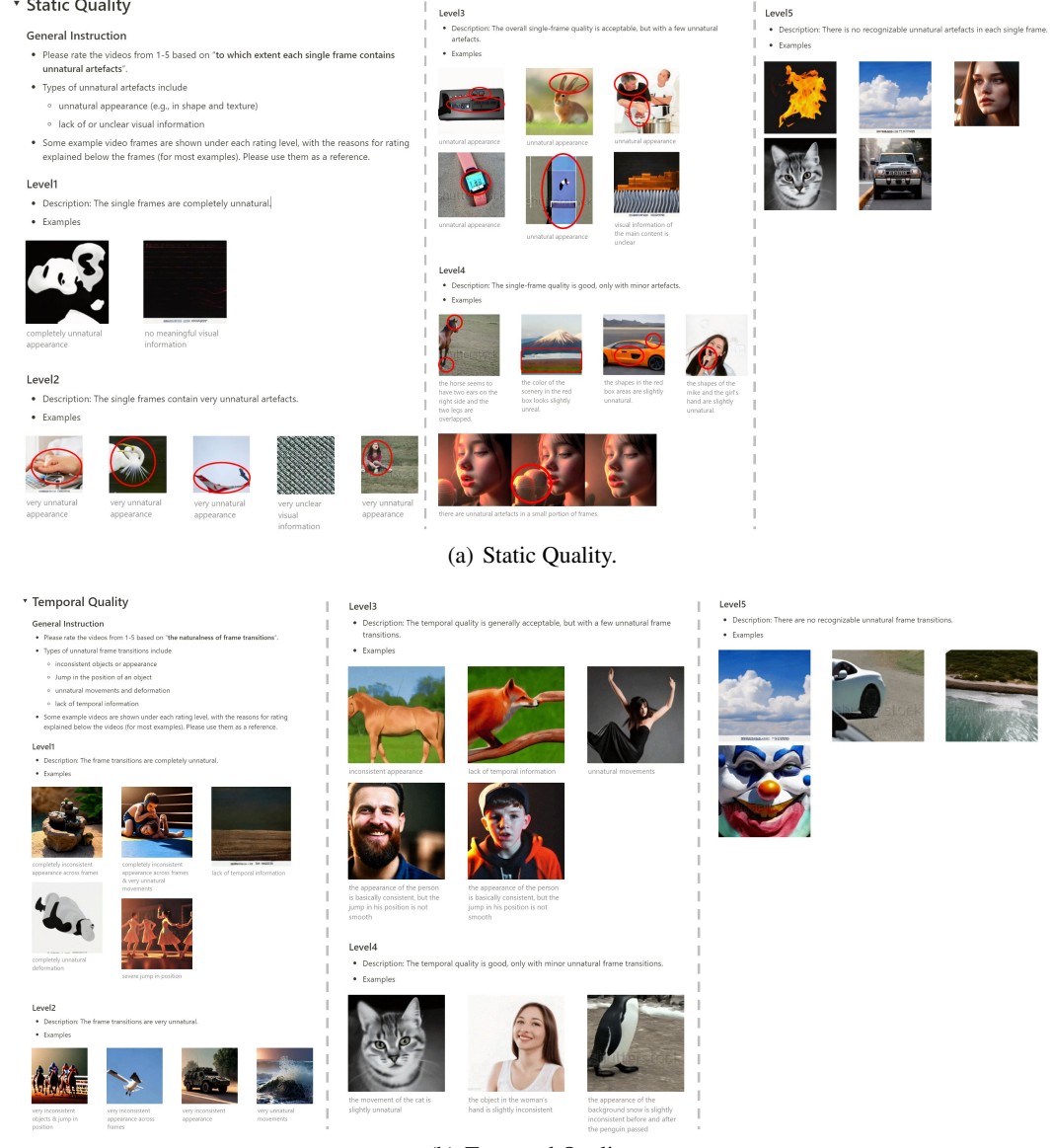

(a) Static Quality.

(b) Temporal Quality.

Figure 14: Screenshots of instructions for manual evaluation of video quality.

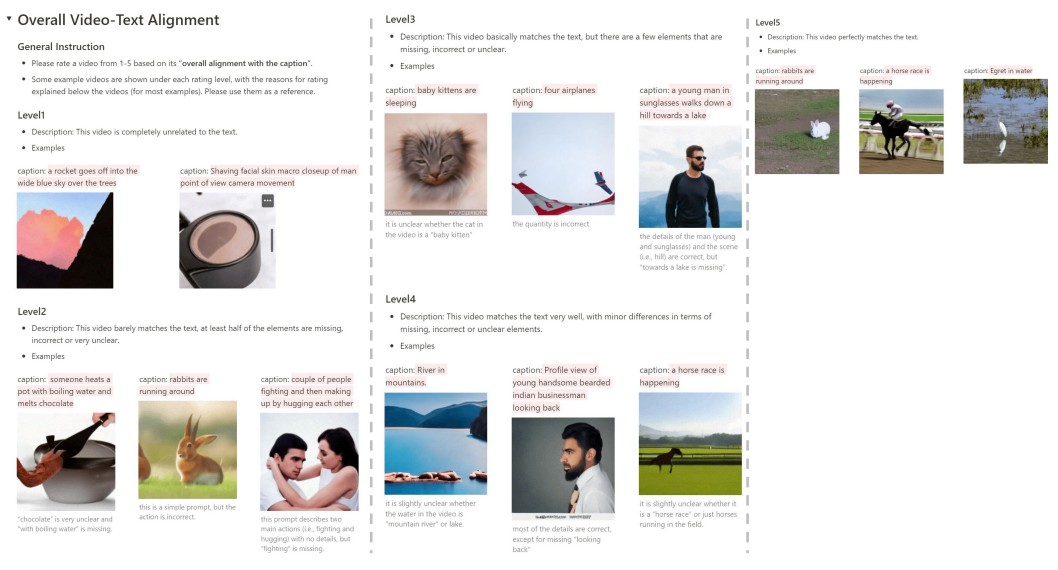

(a) Overall Video-Text Alignment.

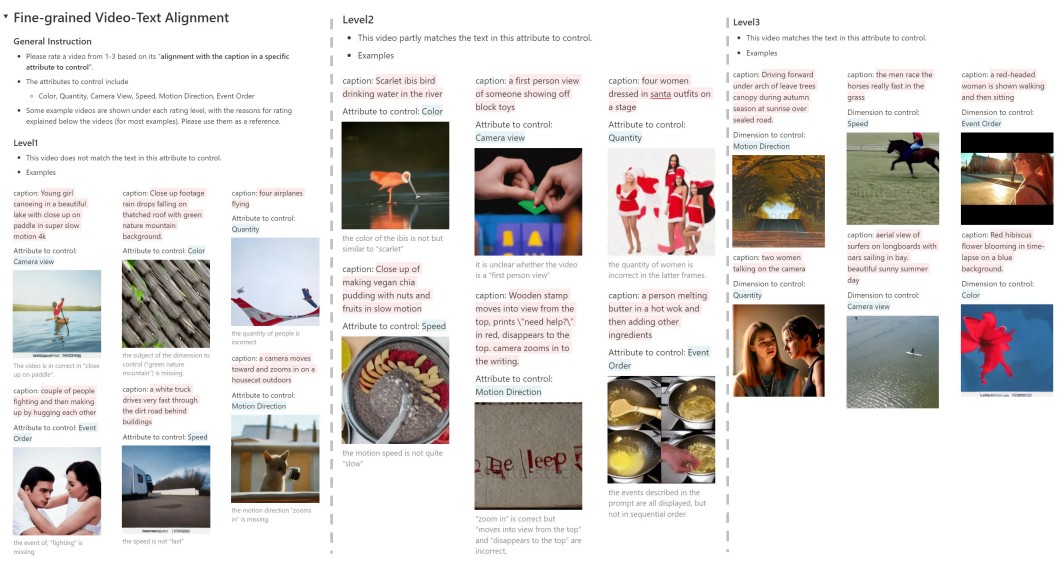

(b) Fine-grained Video-Text Alignment.

Figure 15: Screenshots of instructions for manual evaluation of video-text alignment.

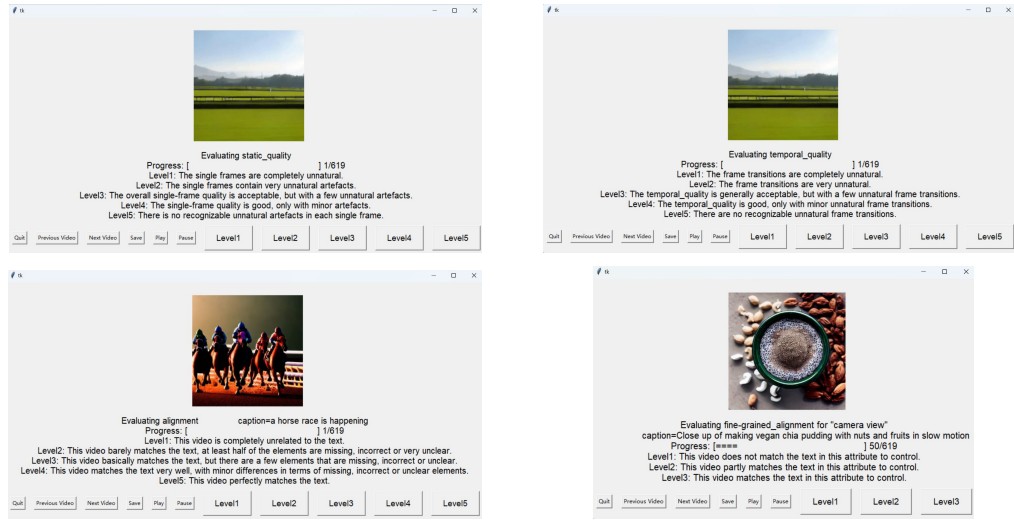

Figure 16: Screenshots of the manual evaluation interface.

more than 20 minutes to generate one video), we only use 300 videos when computing FID/FVD for each category. More reliable results can be obtained by generating more videos.

### F.4 Results on Real-World and Unusual Prompts

Figure 21 compares the models' performance on real-world prompts (collected from existing datasets) and our manually-written unusual prompts. We find that it is more challenging for the models to generate videos following the unusual prompts, while the video quality is less affected.

### F.5 Qualitative Analysis of Generated Videos

Figure 22, 24, 23, 25, 26, 27, 28, 29, 30, 31 present some examples of the generated videos by the three T2V models. We summarize the key findings in the captions, which are in accordance with the findings from the quantitative analysis in Section 4.3. In this appendix, we only show screenshots of the videos. For a better understanding of the video quality and video-text alignment, especially in the temporal categories, please refer to the complete videos.

### F.6 Case study of Video-Text Alignment Metrics

Figure 9 and Figure 32 illustrate examples of videos and video-text alignment score measured by different metrics. We can see that (1) UMTScore produces the most consistent ranking with humans. (2) The other automatic metics are inclined to assign low alignment score to the ground-truth videos. We conjecture that this is because the ground-truth videos are relatively more complicated than the generated ones (e.g., For the prompt "a horse follows two girls" in Figure 32, the two girls do not appear in all video frames.), which requires deep vision-language understanding ability to evaluate.

## G Automatic Categorization Rules

For each category under the "major content" and "attribute control" aspects, we define two types of textual features for matching, i.e., WordNet synsets and key phrases/words, which are summarized in Table 7. Then, the prompts that can match any features of a specific category will be classified accordingly. The "prompt complexity" depends on the number of non-stop words in a prompt, as described in Table 3.

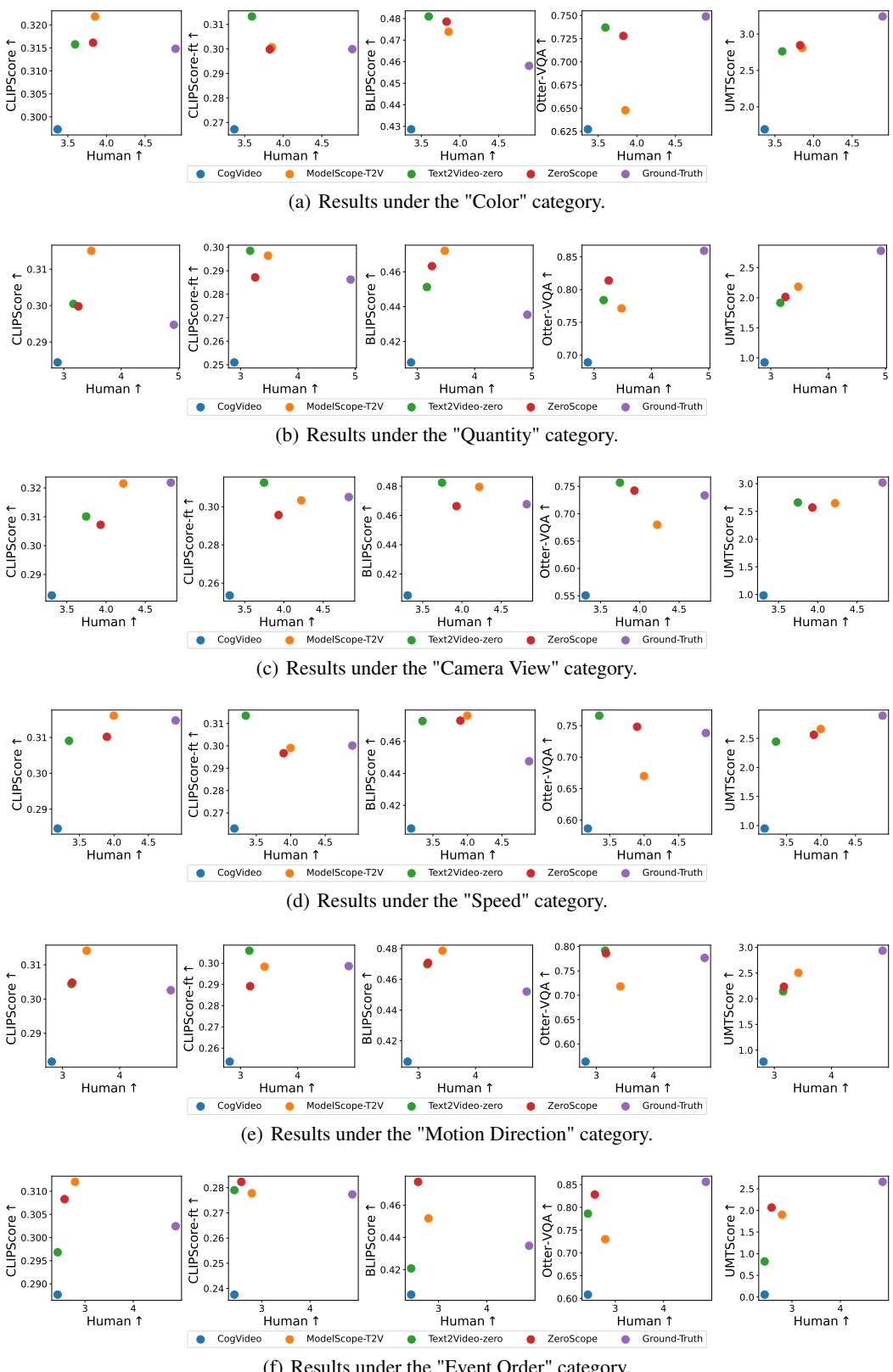

(a) Results under the "Color" category.

(b) Results under the "Quantity" category.

(c) Results under the "Camera View" category.

(d) Results under the "Speed" category.

(e) Results under the "Motion Direction" category.

(f) Results under the "Event Order" category.

Figure 17: Automatic and human ranking of T2V models in terms of overall video-text alignment across different attribute control categories.

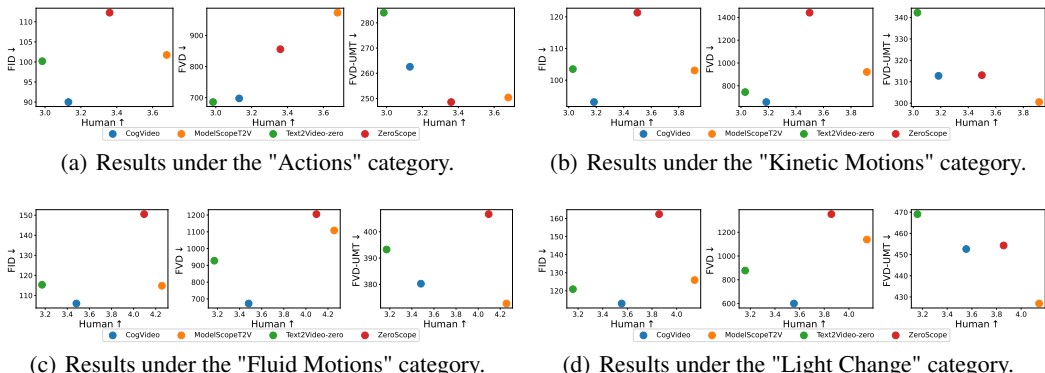

(a) Results under the "Actions" category.

(b) Results under the "Kinetic Motions" category.

(c) Results under the "Fluid Motions" category.

(d) Results under the "Light Change" category.

Figure 18: Automatic and human ranking of T2V models in terms of video quality (averaged static and temporal quality) across different temporal categories.

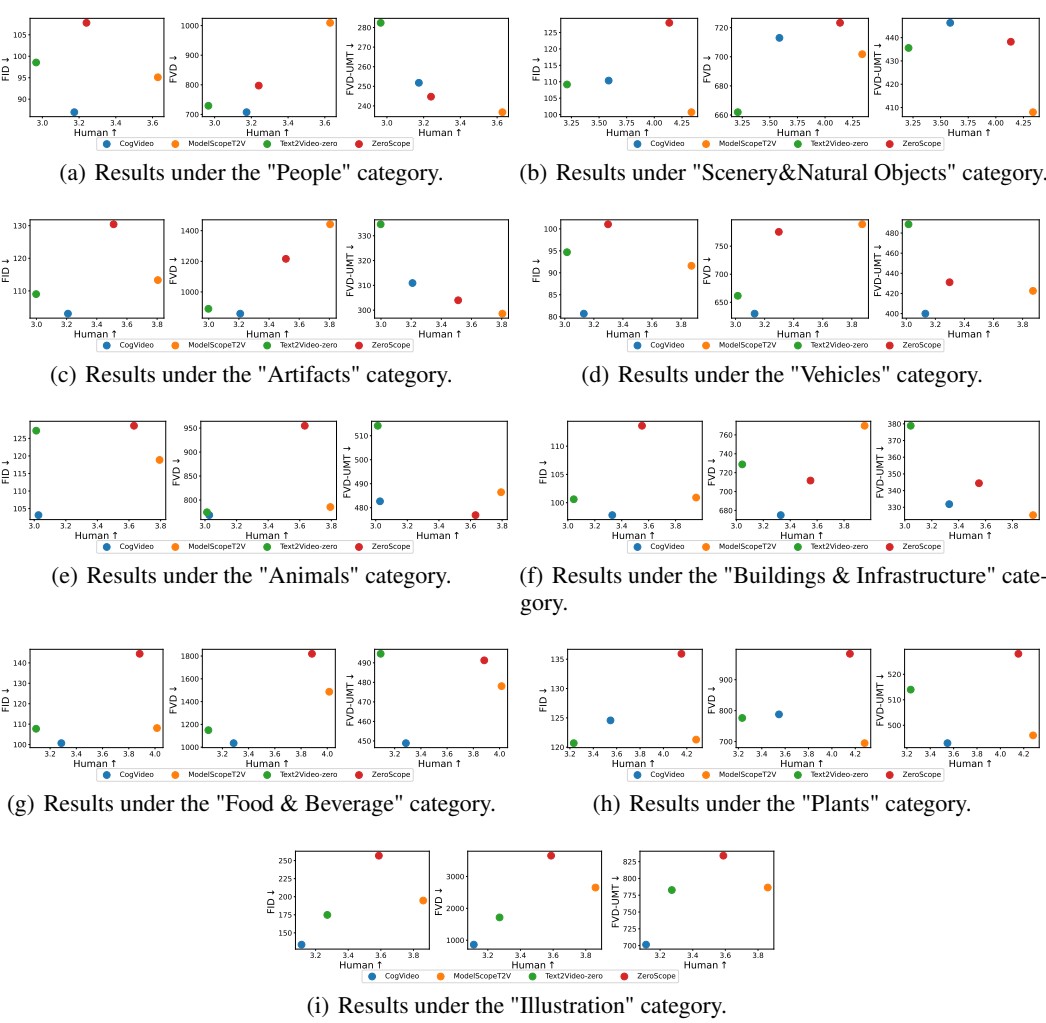

(a) Results under the "People" category.

(b) Results under "Scenery&Natural Objects" category.

(c) Results under the "Artifacts" category.

(d) Results under the "Vehicles" category.

(e) Results under the "Animals" category.

(f) Results under the "Buildings & Infrastructure" category.

(g) Results under the "Food & Beverage" category.

(h) Results under the "Plants" category.

(i) Results under the "Illustration" category.

Figure 19: Automatic and human ranking of T2V models in terms of video quality (averaged static and temporal quality) across different spatial categories.

Table 7: Matching features for automatic categorization.

(a) Temporal categories under the "major content" aspect.

| | WordNet Synsets | Key Phrases/Words |
|---|---|---|
| Actions | 'travel.v.01', 'compete.v.01', 'act.v.01', 'manipulate.v.02', 'move.v.03', 'move.v.02', 'change.v.01', 'make.v.03', 'make.v.01' | None |
| Kinetic Motions | None | 'rotate', 'move', 'bounce', 'spin', 'sway', 'flythrough', 'fly', 'panning', 'drone', 'run', 'walk', 'drive', 'zoom', 'chase', 'swim', 'movement', 'fall', 'rise','sliding video', 'sliding camera', 'sliding shot', 'forward', 'backward', 'leftward', 'rightward', 'upward', 'downward' |
| Fluid Motions | 'body_of_water.n.01', 'fluid.n.01', 'fluid.n.02', 'atmospheric_phenomenon.n.01', 'deformation.n.02' | 'fountain', 'float', 'firework', 'fire', 'cloud', 'clouds', 'candle', 'smoke', 'wave', 'inflate', 'melt', 'shrink', 'ripple' |
| Light Change | 'burning.n.01', 'light.n.01', 'light.n.02', 'light.n.04', 'light.n.07', 'light.n.09' | 'eclipse', 'sunset', 'sunrise', 'firework', 'fire', 'sunbeam', 'sun ray', 'sunshine', 'sunny', 'burn', 'shine', 'luminous', 'glow','explode', 'milky', 'galaxy', 'flash', 'sparkle', 'neon', 'reflection', 'bright', 'candle' |

(b) Spatial categories under the "major content" aspect.

| | WordNet Synsets | Key Phrases/Words |
|---|---|---|
| People | 'person.n.01', 'people.n.01' | 'he', 'she', 'men', 'team' |
| Animals | 'animal.n.01' | None |
| Vehicles | 'vehicle.n.01' | 'drone' |
| Artifacts | 'artifact.n.01' | None |
| Buildings & Infrastructures | 'building.n.01', 'structure.n.01' | 'building', 'cityscape', 'town', 'city' |
| Scenery & Natural Object | 'natural_object.n.01', 'body_of_water.n.01', 'geological_formation.n.01', 'atmospheric_phenomenon.n.01', 'atmosphere.n.05' | 'mountainous', 'fire', 'firework', 'solar eclipse', 'water current', 'water drop', 'cloud', 'desert' |
| Plants | 'plant.n.02', 'vegetation.n.01' | None |
| Food & Beverage | 'food.n.01', 'food.n.02' | 'mushroom' |
| Illustrations | 'shape.n.02', 'symbol.n.01' | 'pattern', 'abstract', 'pattern', 'particle', 'gradient', 'loop', 'graphic' |

(c) Categories under the "attribute control" aspect.

| | WordNet Synsets | Key Phrases/Words |
|---|---|---|
| Color | 'color.n.01', | 'white' |
| Camera View | None | 'macro shot', 'medium shot', 'wide shot', 'close up', 'close-up', 'close view', 'close shot', 'front view', 'front-facing', 'front facing', 'backside view', 'backside shot', 'profile view', 'profile shot', 'side view', 'side shot', 'top view', 'top-down view', 'top down view', 'overhead view', 'overhead shot', 'bottom view', 'bottom shot', 'low angle', 'high angle', 'aerial', 'drone view', "bird's eye view", 'first person', 'first-person', '1st person', 'third person', 'third-person', '3rd person' |
| Quantity | 'integer.n.01' | None |
| Speed | None | 'slow', 'fast', 'slowly', 'fastly', 'time lapse', 'timelapse', 'time-lapse', 'stop motion' |
| Motion Direction | None | 'ahead', 'anticlockwise', 'away from', 'clockwise', 'counterclockwise', 'downward', 'eastbound', 'northbound', 'southbound', 'westbound', 'homeward', 'leftwards', 'rightwards', 'upward', 'left', 'right', 'forward', 'backward', 'toward', 'out of', 'approach', 'leave', 'against to', 'lift', 'opposite direction' |
| Event Order | None | 'and then' |

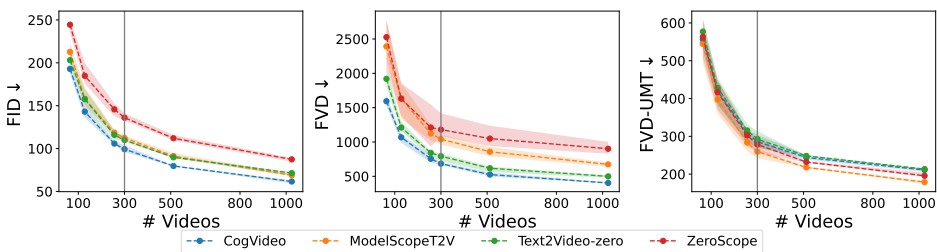

Figure 20: The effect of video sample number on FID and FVD. The shaded regions represent the highest and lowest values observed across the four runs. The human evaluation of video quality (average of static and temporal quality) are 3.08, 3.27, 3.62 and 3.93 for Text2Video-zero, CogVideo, ZeroScope and ModelScopeT2V, respectively.

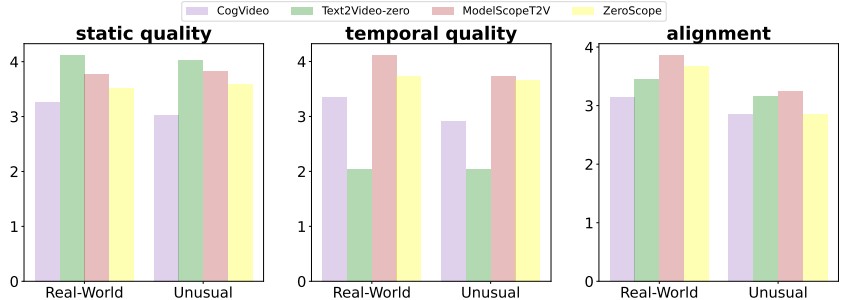

Figure 21: Manual evaluation results on real-world and unusual prompts.

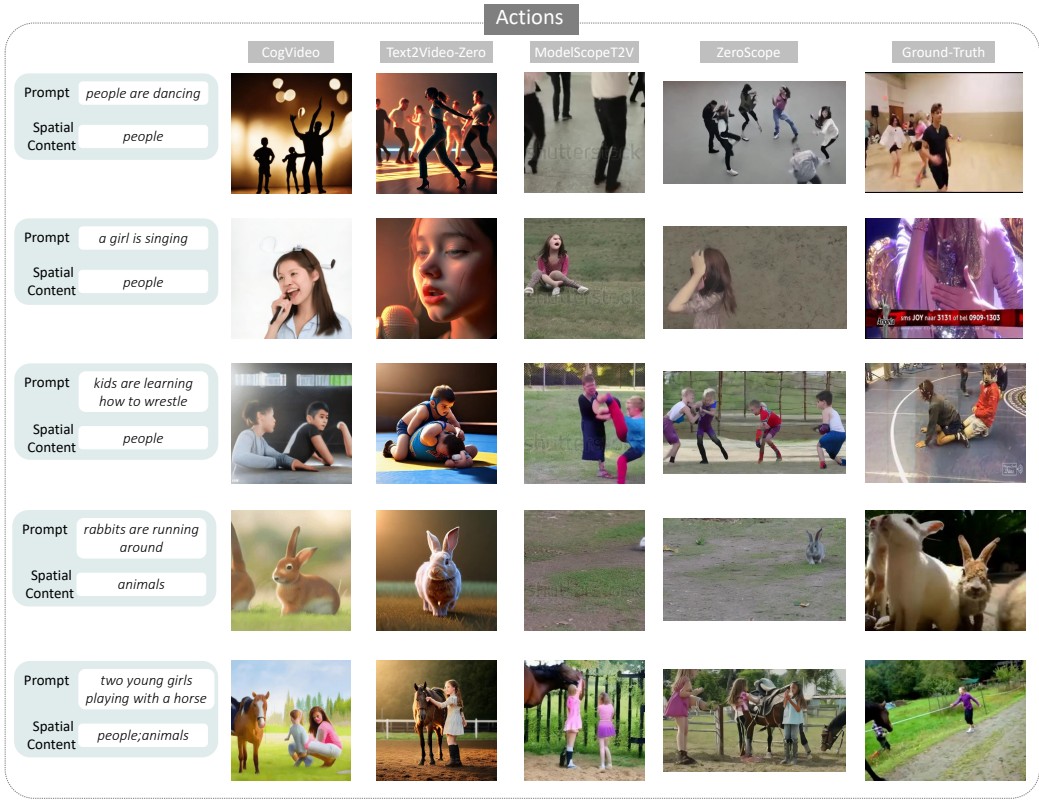

Figure 22: Screenshots of videos generated from prompts of the "Actions" category. Most generated videos contain unnatural elements, including unnatural appearances, inconsistent appearances across frames, unnatural movements and jump in positions.

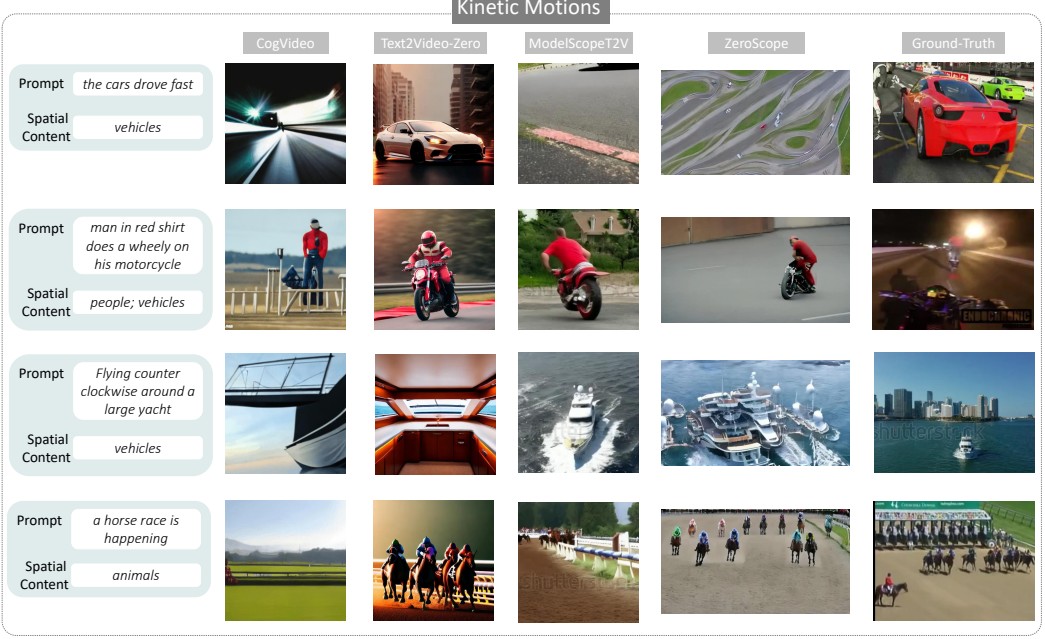

Figure 23: Screenshots of videos generated from prompts of the "Kinetic Motions" category. Most generated videos contain unnatural elements, including unnatural appearances, inconsistent appearances across frames, unnatural movements and jump in positions.

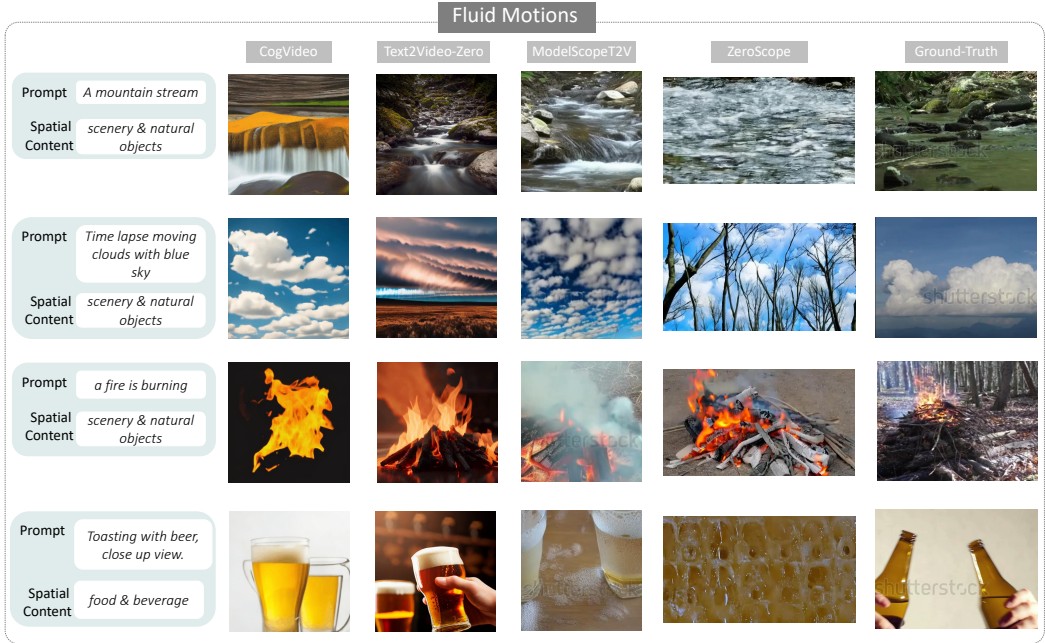

Figure 24: Screenshots of videos generated from prompts of the "Fluid Motions" category. Most generated videos are of good quality, except for Text2Video-Zero, which exhibits poor temporal quality.

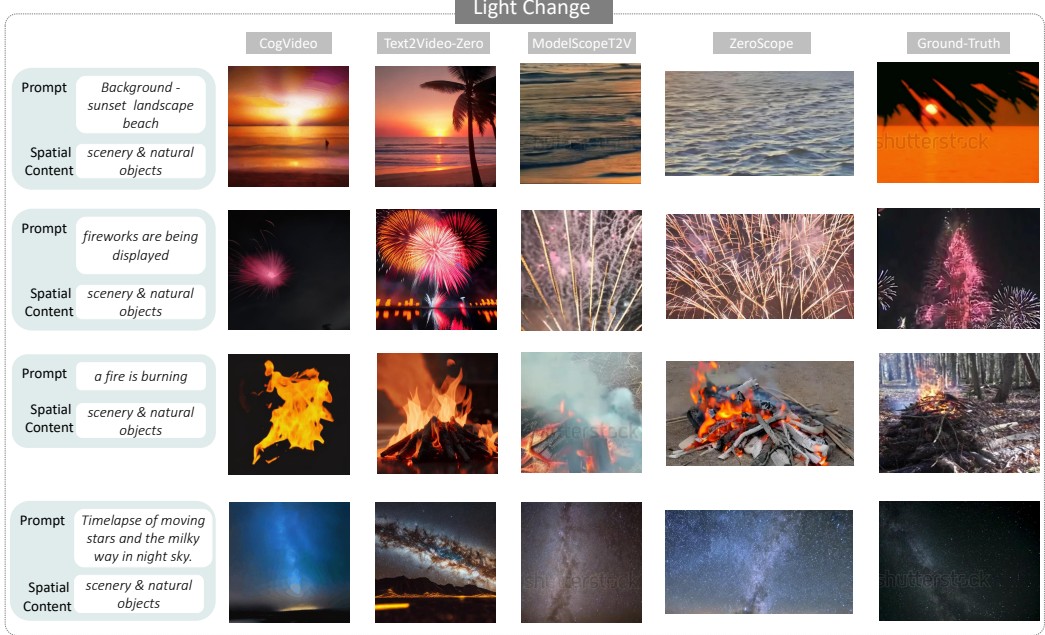

Figure 25: Screenshots of videos generated from prompts of the "Light Change" category. Most generated videos are of good quality, except for Text2Video-Zero, which exhibits poor temporal quality.

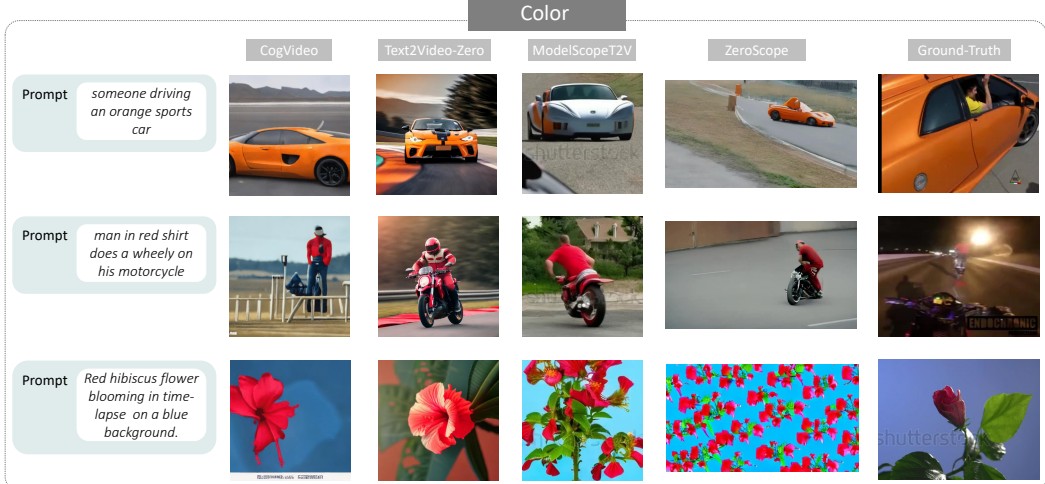

Figure 26: Screenshots of videos generated from prompts of the "Color" category. All T2V models can control color in most cases.

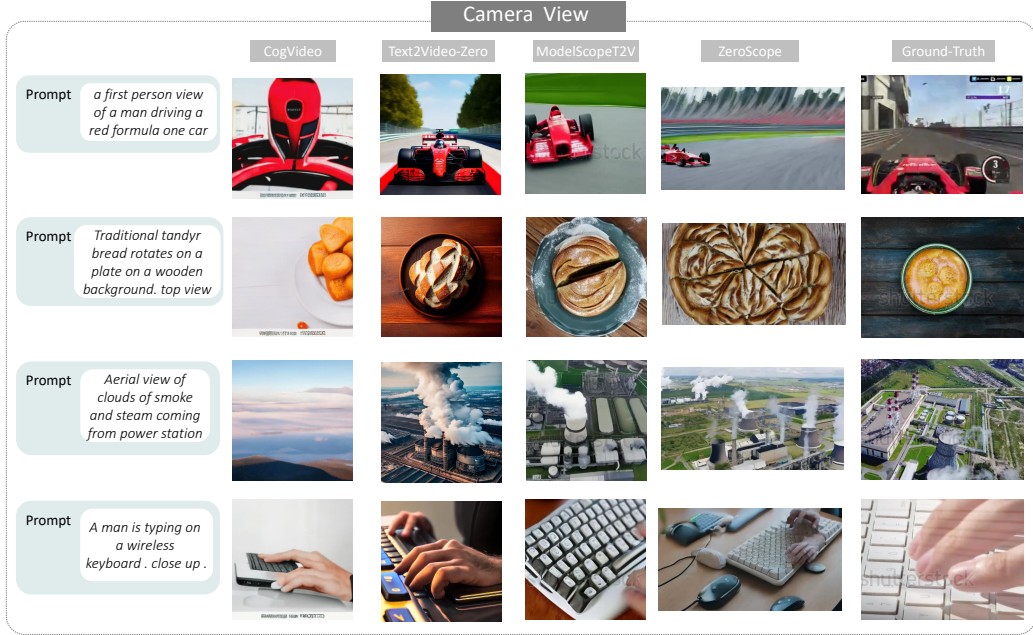

Figure 27: Screenshots of videos generated from prompts of the "Camera View" category. All T2V models can control camera view in most cases.

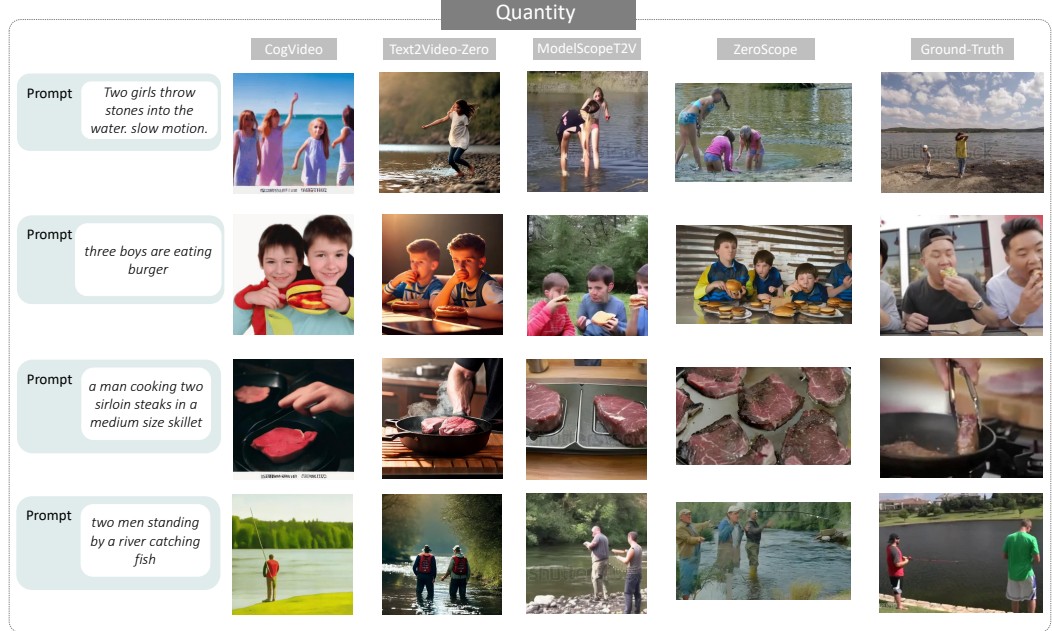

Figure 28: Screenshots of videos generated from prompts of the "Quantity" category. ModelScopeT2V and Text2Video-Zero can control the quantity of objects to some extent, while CogVideo fails in most cases.

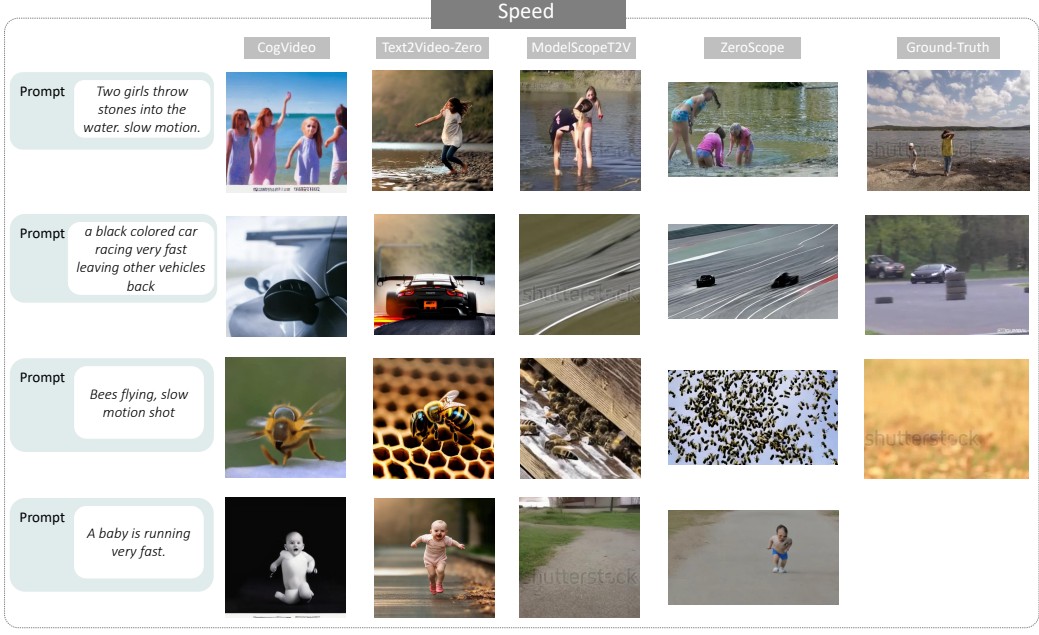

Figure 29: Screenshots of videos generated from prompts of the "Speed" category. ModelScopeT2V can control both fast and slow speeds to some extent. CogVideo can generate some videos with "slow motion" while struggling with fast speed. For Text2Video-Zero, speed can only be reflected via static information (e.g., the posture of the baby in the last case).

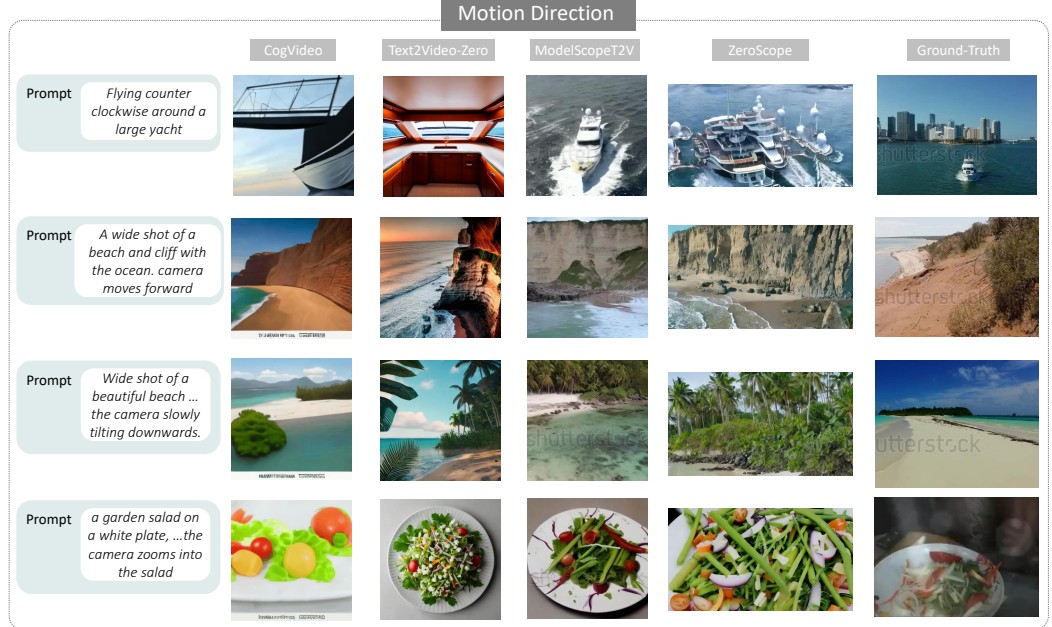

Figure 30: Screenshots of videos generated from prompts of the "Motion Direction" category. ModelScopeT2V and ZeroScope can control motion direction to some extent, while CogVideo and Text2Video-Zero fail in most cases.

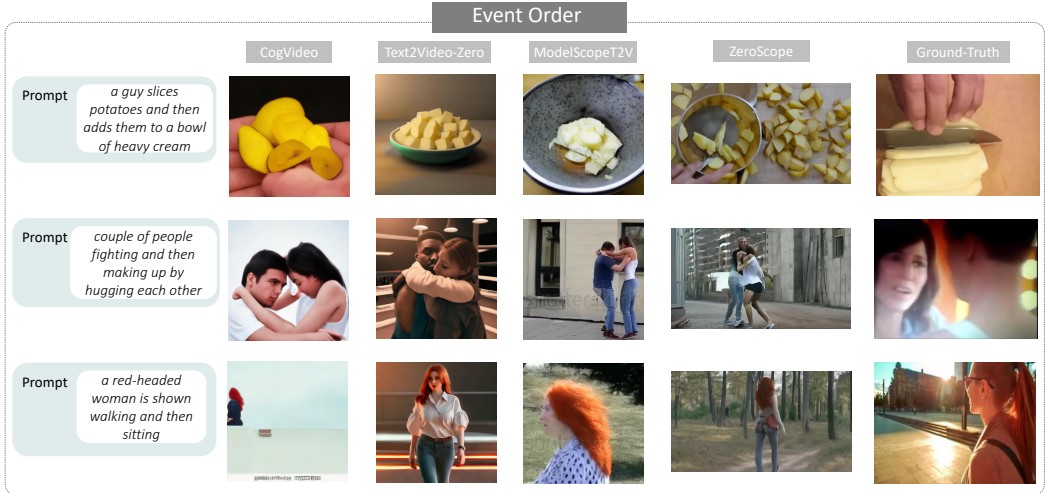

Figure 31: Screenshots of videos generated from prompts of the "Event Order" category. All T2V models cannot generate videos according to the given event order.

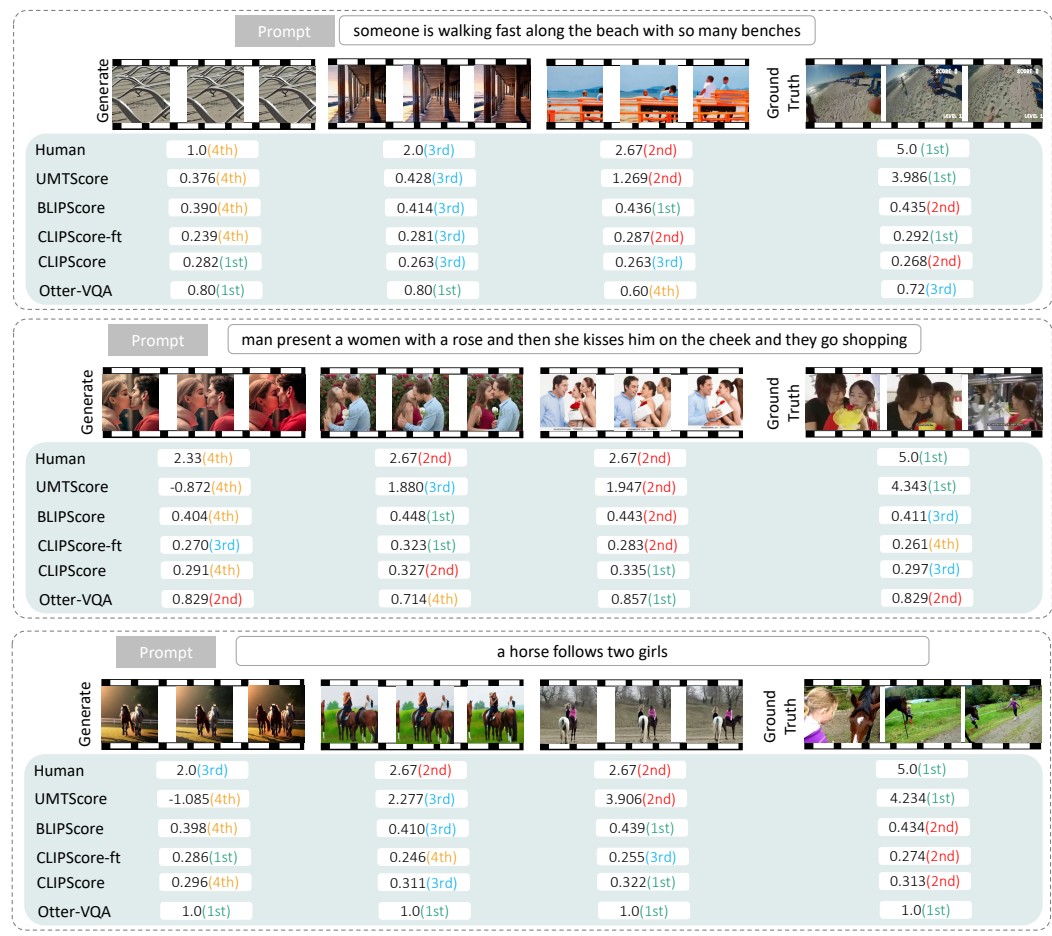

Figure 32: Video examples and alignment scores measured by different metrics. The left three videos are generated and the rightmost video is the ground-truth.

# H  Licensing, Hosting and Maintenance Plan

**Author Statement.**   We bear all responsibilities for the licensing, distribution, and maintenance of our dataset.

**License.**   FETV is under CC-BY 4.0 license.

**Hosting.**   FETV can be viewed and downloaded on GitHub at `https://github.com/llyx97/FETV` or on Huggingface at `https://huggingface.co/datasets/lyx97/FETV`, and will be preserved for a long time. The dataset is in the JSON file format.

**Metadata.**   Metadata can be found at `https://huggingface.co/datasets/lyx97/FETV`.

# I  Datasheet

## I.1  Motivation

**For what purpose was the dataset created?**

**Answer:**   FETV is created to facilitate fine-grained evaluation in the task of open-domain text-to-video generation. Compared with previous benchmarks for this task, FETV provides a more comprehensive view of T2V models' capabilities from multiple aspects.

**Who created the dataset (e.g., which team, research group) and on behalf of which entity (e.g., company, institution, organization)?**

**Answer:**   This evaluation dataset is created by Yuanxin Liu , Lei Li, Shuhuai Ren, Rundong Gao, Shicheng Li, Sishuo Chen and Xu Sun (Peking University) and Lu Hou (Huawei Noah's Ark Lab).

**Who funded the creation of the dataset?**

**Answer:**   Peking University and Huawei Technologies Co., Ltd.

## I.2  Composition

**What do the instances that comprise the dataset represent? (e.g., documents, photos, people, countries)**

**Answer:**   Our data is provided in JSON format. Each data instance consists of (1) a text prompt, (2) the categorization labels under three aspects, (3) the source of the prompt (collected from MSR-VTT and WebVid or manually written by us), (4) ID and (5) URL link of the reference video.

**How many instances are there in total (of each type, if appropriate)?**

**Answer:**   In total, we collect 619 instances. The data distribution over different categories can be found in Figure 3 and Figure 4 of the main paper.

**Does the dataset contain all possible instances or is it a sample (not necessarily random) of instances from a larger set?**

**Answer:**   A part of the text prompts in FETV is collected from existing datasets of open-domain text-video pairs (i.e., MSR-VTT and WebVid). When collecting prompts, we ensure that each category in our proposed categorization system has sufficient prompts.

**Is there a label or target associated with each instance?**

**Answer:**   Each prompt collected from MSR-VTT and WebVid is associated with a reference video, which is provided in the form of an URL link.

**Is any information missing from individual instances?**

**Answer:**   For the manually written unusual prompts, FETV does not provide the URL link to reference videos.

**Are relationships between individual instances made explicit (e.g., users' movie ratings, social network links)?**

**Answer:**   The text prompts in FETV belong to different categories, which can be seen from the categorization labels.

**Are there recommended data splits (e.g., training, development/validation, testing)?**

**Answer:** No. The entire FETV benchmark is intended for evaluation.

**Are there any errors, sources of noise, or redundancies in the dataset?**

**Answer:** No.

**Is the dataset self-contained, or does it link to or otherwise rely on external resources (e.g., websites, tweets, other datasets)?**

**Answer:** The reference videos are provided as URL links because we do not own the copyright to the videos.

**Does the dataset contain data that might be considered confidential?**

**Answer:** No.

**Does the dataset contain data that, if viewed directly, might be offensive, insulting, threatening, or might otherwise cause anxiety?**

**Answer:** No.

### I.3 Collection Process

The data collection process is described in Section 3.2 of the main paper and Appendix G.

### I.4 Uses

**Has the dataset been used for any tasks already?**

**Answer:** Yes. We have used the FETV benchmark to evaluate three existing text-to-video generation models and diagnose the reliability of automatic video quality and video-text alignment metrics. Please see Section 4 and Section 5 of the main paper for details.

**What (other) tasks could the dataset be used for?**

**Answer:** FETV is mainly designed to evaluate open-domain text-to-video generation models, but it can also be extended to test the reliability of automatic evaluation metrics, by computing their correlation between human judgements (as shown in Section 4).

**Is there a repository that links to any or all papers or systems that use the dataset?**

**Answer:** No.

**Is there anything about the composition of the dataset or the way it was collected and preprocessed/cleaned/labeled that might impact future uses?**

**Answer:** In the first stage of our data collection process, some prompts that actually belong to a particular category might be filtered out by the automatic matching rules (introduced in Appendix G), which potentially impacts the language diversity of the prompts.

**Are there tasks for which the dataset should not be used?**

**Answer:** FETV mainly consists of sentence-level text prompts, and thus it is not suitable for evaluating video generation based on paragraph-level texts.

### I.5 Distribution

**Will the dataset be distributed to third parties outside of the entity (e.g., company, institution, organization) on behalf of which the dataset was created?**

**Answer:** Yes. The benchmark is publicly available on the Internet.

**How will the dataset will be distributed (e.g., tarball on website, API, GitHub)?**

**Answer:** The benchmark is available on GitHub at `https://github.com/llyx97/FETV` or on Huggingface at `https://huggingface.co/datasets/lyx97/FETV`.

**Will the dataset be distributed under a copyright or other intellectual property (IP) license, and/or under applicable terms of use (ToU)?**

**Answer:** CC-By 4.0.

**Have any third parties imposed IP-based or other restrictions on the data associated with the instances?**

**Answer:** No.

**Do any export controls or other regulatory restrictions apply to the dataset or to individual instances?**

**Answer:** No.

### I.6 Maintenance

**Who will be supporting/hosting/maintaining the dataset?**

**Answer:** The authors will be supporting, hosting, and maintaining the dataset.

**How can the owner/curator/manager of the dataset be contacted (e.g., email address)?**

**Answer:** Please contact Yuanxin Liu (liuyuanxin@stu.pku.edu.cn).

**Is there an erratum?**

**Answer:** No. We will make announcements if there are any.

**Will the dataset be updated (e.g., to correct labeling errors, add new instances, delete instances)?**

**Answer:** Yes. We will post new update in `https://huggingface.co/datasets/lyx97/FETV` and `https://github.com/llyx97/FETV` if there is any.

**If the dataset relates to people, are there applicable limits on the retention of the data associated with the instances (e.g., were individuals in question told that their data would be retained for a fixed period of time and then deleted)?**

**Answer:** People may appear in the reference videos. People may contact us to exclude specific data instances if they appear in the reference videos.

**Will older versions of the dataset continue to be supported/hosted/maintained?**

**Answer:** Yes. Old versions will also be hosted in `https://huggingface.co/datasets/lyx97/FETV` and `https://github.com/llyx97/FETV`.

**If others want to extend/augment/build on/contribute to the dataset, is there a mechanism for them to do so?**

**Answer:** Yes. Others can extend FETV to collect more prompts for evaluation, based on our multi-aspect categorization system (introduced in Section 3.1 of the main paper).

