# OpenReview forum: "FETV: A Benchmark for Fine-Grained Evaluation of Open-Domain Text-to-Video Generation"
_NeurIPS.cc/2023/Track/Datasets_and_Benchmarks — NeurIPS 2023 Datasets and Benchmarks Poster_

### Official Review · Reviewer_5H2c · 2023-07-20
**Comments of FETV**

**Rating:** 6
**Confidence:** 5
**Correctness:** Yes.
**Clarity:** Yes.

**Strengths:**

- S1. The proposed FETV includes evaluation from various aspects, which is more comprehensive than other related benchmarks.
- S2. The authors succeed in pointing out the limitation of automatic evaluation.

**Additional Feedback:**

- Concern about automatic evaluation. I agree with your opinion that solid quantitative metrics are critical for Text-I/V generation tasks. These metrics should correctly measure the content relevance of the generated I/V and the temporal consistency between the frames. However, this paper fails to provide some solid metrics. Recently, various Vision Language Large Models (VLLM) are proposed. I'd like to know if is it possible to use Vision Question Answering or Video Caption metrics to measure the content relevance. Some VLLM models (Otter, BLIPv2, EMU) have shown strong VQA and VC abilities.

- Since temporal consistency is an important aspect of video generation tasks, a comprehensive T2IV benchmark should measure the temporal consistency between frames. The authors should take the temporal consistency into consideration and better justify their benchmark.

- More Text-Video generation methods should be evaluated. Three models are insufficient.

- Following are some questions:

-- Q1. How did you get the ground truths for unusual prompts?

-- Q2. In Table. 2, the human performance is also poor. Does this indicate the low-quality of the collected dataset?

Suggestion:

In Figure. 5, some texts are occlusion by the images.

The authors should clearly address my concerns on W1 and W2. I will update my final rating according to the rebuttal.

References:

Emu: https://github.com/baaivision/Emu

Otter: https://github.com/Luodian/Otter

**Documentation:**

Yes.

**Ethics:**

No.

**Limitations:**

As mentioned above, there are two major limitations from my point of view, which are the failure of finding an **automatic evaluation** and the insufficient discussion on **temporal consistency**. I have justified my concerns in the Additional Feedback section in detail.

**Opportunities For Improvement:**

- W1. This work fails to provide a solid metric for automatic performance evaluation.
- W2. Temporal consistency is an important aspect of video generation tasks. Discontinuous textures and actions between frames can damage the viewing experience. A video generation benchmark should model temporal consistency. The authors should pay more attention to discussing temporal consistency.
- W3. Some introductions are vague.

**Relation To Prior Work:**

Yes.

**Summary And Contributions:**

This paper collects and builds a new evaluation benchmark FETV for Text-to-Video generation. The proposed FETV contains around 600 prompts and 3 orthogonal aspects, which are major content, attributes to control, and prompt complexity. Some experiments show the weaknesses of current automatic metrics. Therefore, the FETV should be evaluated manually.

---

> ### Author Response · Authors · 2023-08-26
>
> ### 1. Solid metric for automatic performance evaluation
> ### Reviewer Comment:
>     This work fails to provide a solid metric for automatic performance evaluation.
>
>     I agree with your opinion that solid quantitative metrics are critical for Text-I/V generation tasks. These metrics should correctly measure the content relevance of the generated I/V and the temporal consistency between the frames. However, this paper fails to provide some solid metrics.
> ### Response
> We fully recognize the importance of reliable automatic metrics in evaluating and developing T2V generation models. To obtain a solid automatic metric, we employed an advanced video-language model, namely UMT [1], to evaluate the video-text alignment. We find that the UMT-based metric is consistent with the human ranking of T2V models. Please refer to the general response for details.
>
> ## 2. Measure and discussion of temporal consistency
> ### Reviewer Comment
>     Since temporal consistency is an important aspect of video generation tasks, a comprehensive T2IV benchmark should measure the temporal consistency between frames. The authors should take the temporal consistency into consideration and better justify their benchmark.
>
>     Temporal consistency is an important aspect of video generation tasks. Discontinuous textures and actions between frames can damage the viewing experience. A video generation benchmark should model temporal consistency. The authors should pay more attention to discussing temporal consistency.
> ### Response
> We agree that temporal consistency is important for video generation and, in fact, we have considered this aspect in our evaluation. We refer to it as "temporal quality" as described in Section 3.2 (line 131). Our "temporal major content" aspect also considers temporal consistency by introducing four types of temporal content, i.e., actions, kinetic motions, fluid motions and light change. Our evaluation results suggest that existing T2V generation models struggle with temporal consistency of actions and kinetic motions and perform relatively well in fluid motions and light change.
>
> ## 3. Vision LLM-based Metrics
> ### Reviewer Comment
>     I'd like to know if is it possible to use Vision Question Answering or Video Caption metrics to measure the content relevance. Some VLLM models (Otter, BLIPv2, EMU) have shown strong VQA and VC abilities.
> ### Response
>   Thanks for the constructive suggestion! We have adopted two Video LLMs (Otter [2] and VideoChat [3]) to evaluate video-text alignment by formulating it as a Video QA task.
>
>   Specifically, we first extract the key elements from the text prompt and generate corresponding yes-no questions via the Vicuna model [4]. For instance, given the prompt "people are dancing", the extracted elements are "people" and "dancing", and the question for "people" is "are there people in the video?". Then, we feed the generated (or ground-truth) videos and questions into the Video LLM. The average number of questions that receive a positive answer is defined as the alignment score for a specific video-text pair. We name these metrics as VideoChat-VQA and Otter-VQA, respectively.
>
>   The correlation with human evaluation is reported below. We can see that the performance of VideoChat-VQA and Otter-VQA is clearly worse than BLIPScore and CLIPScore. We find that this is because VideoChat and Otter cannot acurately answer the questions based on the videos (The generated questions and Video LLM's answers can be found in https://github.com/llyx97/FETV/tree/main/video_qa_eval). This suggests that the video-text understanding ability of existing Video LLMs is still insufficient to provide a reliable evaluation of T2V generation. The detailed implementation of VideoChat-VQA and Otter-VQA and analysis of the results will be provided in the next version of our paper.
>
> | Metrics | Kendall | Spearman |
> | --- | --- | --- |
> | CLIPScore-ft | 23.3 | 32.3 |
> | BLIPScore | 24.8 | 34.0 |
> | VideoChat-VQA | 12.6 | 19.1 |
> | Otter-VQA | 10.3 | 14.5 |
> | Inter-Human | 58.0 | 73.0 |
>
> ## 4. More Text-Video generation methods
> ### Reviewer Comment
>     More Text-Video generation methods should be evaluated. Three models are insufficient.
> ### Response
> We can fully understand this concern. In response to the request, we are evaluating a more recent model ZeroScope [5] and the results will be reported in the next version of our paper. We would also like to note that the three models we evaluated were the only ones available as open source at the writing of the paper. Please refer to the general response for more details.

---

> > ### Author Response · Authors · 2023-08-26
> >
> > ## 5. How did you get the ground truths for unusual prompts?
> > ### Response
> >   As we described in the implementation details, we evaluate the 541 reference videos (line 144). These videos correspond to real-world prompts and we do not consider ground-truth videos for the 78 unusual prompts because it is difficult to find such videos. We will make this detail clearer in the revised paper.
> >
> > ## 6. Human performance in Table. 2
> > ### Reviewer Comment
> >     In Table. 2, the human performance is also poor. Does this indicate the low-quality of the collected dataset?
> > ### Response:
> >   Thanks for raising this question. In fact, the correlation between our human evaluators is strong. In the evaluation of alignment between video and machine-generated captions [6], the Kendall and Spearman coefficients are 0.568 and 0.628 respectively, on a 1-5 Likert scale. In comparison, the Kendall and Spearman coefficients for our human evaluation of video-text alignment (1-5 Likert-scale) are 0.580 and 0.730. In [7], the inter-annotator agreement (IAA) measured by Krippendorff’s α for image-text alignment ratings is 0.48 (1-5 Likert-scale), while ours is 0.638. We will clarify this question in the revised paper and add the IAA results.
> >
> > ## 7. Occlusion In Figure. 5
> > ### Reviewer Comment
> >     Some texts are occlusion by the images
> > ### Response:
> > Thanks for the feedback. We will resolve this issue in the next version.
> >
> > [1] Unmasked Teacher: Towards Training-Efficient Video Foundation Models. ICCV23.
> >
> > [2] Otter: A Multi-Modal Model with In-Context Instruction Tuning.
> >
> > [3] VideoChat: Chat-Centric Video Understanding.
> >
> > [4] https://huggingface.co/lmsys/vicuna-13b-v1.3
> >
> > [5] https://huggingface.co/cerspense/zeroscope_v2_576w
> >
> > [6] EMScore: Evaluating Video Captioning via Coarse-Grained and Fine-Grained Embedding Matching. CVPR22.
> >
> > [7] Toward Verifiable and Reproducible Human Evaluation for Text-to-Image Generation. CVPR23.

---

> > > ### Comment · Reviewer_5H2c · 2023-08-26
> > > **Response to the rebuttal**
> > >
> > > Hi authors,
> > > Thank you for providing a detailed response. I'm glad to see a solid automatic metric can be proposed. My major concerns are well addressed. I still have some additional concerns:
> > >
> > > C1. A benchmark paper should be **comprehensive**. The authors should re-implement and evaluate more related methods. Although the authors have implemented recent SOTA methods, the number of methods included in the paper is still limited. Please include more typical methods (e.g., 6~8 methods in total).
> > >
> > > C2. The authors have demonstrated the quality of the proposed dataset. Can you provide more details to show the high quality of your dataset? As an evaluation dataset, its quality should be very high.
> > >
> > > C3. As a video benchmark, providing some cases in the supplementary material will be very helpful. Can you pick some cases to highlight the weaknesses of related methods and highlight the diversity and quality of the proposed dataset?
> > >
> > > I will raise my score based on further justification and paper revision.

---

> > > > ### Author Response · Authors · 2023-08-29
> > > > **Further Response to Reviewer 5H2c**
> > > >
> > > > Thank you so much for taking the time to read our rebuttal and providing a prompt response! We reply to the further concerns as follows:
> > > >
> > > > ## C.1 Evaluation of more methods
> > > > We agree that a benchmark paper should comprehensively evaluate related methods and we have done our best to evaluate all the open-source models available. In the following table, we summarize the open T2V models published before the abstract submission deadline (2023.6.1). We have evaluated the three open-sourced models, i.e., CogVideo, Text2Video-zero and VideoFusion. The results of the more recent model, ZeroScope, are also reported in the revised paper. Additionally, with comprehensive information about our human evaluation protocol and the released automatic evaluation code, we believe that it would be easy for follow-up researchers to evaluate future T2V models.
> > > > | Model | Is Open-Sourced | Year | Link |
> > > > | --- | --- | --- | --- |
> > > > | CogVideo | ✓ | 2022 | https://arxiv.org/abs/2205.15868 |
> > > > | Text2Video-zero | ✓ | 2023.3 | https://arxiv.org/abs/2303.13439 |
> > > > | VideoFusion | ✓ | 2023.3 | https://arxiv.org/abs/2303.08320 |
> > > > | ZeroScope | ✓ | 2023.6 | https://huggingface.co/cerspense/zeroscope_v2_576w |
> > > > | NUWA | x | 2021 | https://arxiv.org/abs/2111.12417 |
> > > > | GODIVA | x | 2021 | https://arxiv.org/abs/2104.14806 |
> > > > | Video Diffusion Model | x | 2022 | https://arxiv.org/abs/2204.03458 |
> > > > | Make-a-Video | x | 2022 | https://arxiv.org/abs/2209.14792 |
> > > > |  Phenaki | x | 2022 | https://arxiv.org/abs/2210.02399 |
> > > > | Imagen Video | x | 2022 | https://arxiv.org/abs/2210.02303 |
> > > > | Magic Video | x | 2022 | https://arxiv.org/abs/2211.11018 |
> > > > | VideoLDM | x | 2023.4 | https://arxiv.org/abs/2304.08818 |
> > > >
> > > > ## C.2 More details of dataset quality
> > > >   Here we provide additional details from the category distribution perspective to illustrate the advantage of FETV over previous video-text datasets as a benchmark for open T2V. To obtain the distribution of MSR-VTT and WebVid, we randomly sample 100 data from each dataset and manually annotate the categories.
> > > >
> > > >   As we can see in the following results, FETV exhibits a more uniform distribution over the categories, while MSR-VTT and WebVid are more biased toward certain categories. Particularly, more than 60\% of the MSR-VTT and WebVid prompts do not involve the six attributes introduced in FETV. Consequently, the ability to control these categories, especially the challenging ones, cannot be reflected by MSR-VTT and WebVid (note that these two datasets lack categorization of the attributes).
> > > >
> > > > These results and discussions are also provided in the newly updated paper appendix.
> > > >
> > > > ### Distribution over attributes to control
> > > >
> > > > |  | Color | Camera View | Quantity | Speed | Motion Direction | Event Order | None |
> > > > | :---: | :---: | :---: | :---: | :---: | :---: | :---: | :---: |
> > > > | MSR-VTT+WebVid | 15.5% | 10% | 5% | 6% | 9% | 1% | 63% |
> > > > | FETV | 22.3% | 14.9% | 10.3% | 18.3% | 15.3% | 12.0% | 26.8% |
> > > >
> > > > ### Distribution over spatial major contents
> > > >
> > > > |  | People | Animals | Vehicles | Artifacts | Plants | Buildings | Food | Scenery | Illustrations | None |
> > > > | :---: | :---: | :---: | :---: | :---: | :---: | :---: | :---: | :---: | :---: | :---: |
> > > > | MSR-VTT+WebVid | 61% | 5.5% | 12.5% | 33% | 8% | 19.5% | 5% | 22% | 1.5% | 4.5% |
> > > > | FETV | 47.7% | 17% | 21.2% | 25.5% | 8.6% | 14.9% | 12.4% | 27% | 3.1% | 0% |
> > > >
> > > > ### Distribution over temporal major contents
> > > >
> > > > |  | Actions | Kinetic Motoins | Fluid Motions | Light Change | None |
> > > > | :---: | :---: | :---: | :---: | :---: | :---: |
> > > > | MSR-VTT+WebVid | 64.5% | 23% | 14.5% | 6.5% | 12.5% |
> > > > | FETV | 61.4% | 49.3% | 34.7% | 15.5% | 0% |
> > > >
> > > > ## C.3 More case studies
> > > > Thanks for the constructive suggestion! In fact, we have presented a case study of the generated videos from different categories in Appendix F.1 (Qualitative Analysis of Generated Videos). To provide a more intuitive sense of the performance of generated videos, we summarize the videos, together with the prompt, category and human rating scores, in our dataset repo (https://github.com/llyx97/FETV/tree/main/generated_video_examples).

---

> > > > > ### Comment · Reviewer_5H2c · 2023-08-30
> > > > >
> > > > > Hi authors,
> > > > >
> > > > > My concerns have been well addressed. It will be time-consuming but worthwhile for the authors to re-implement more SOTA methods and the authors have included methods as many as possible. I also have checked the visual videos. The temporal abilities of these methods are catastrophic. This work has the value to guide related future works. Ergo, I decide to raise my score from 5 to 6.

---

### Official Review · Reviewer_2CxJ · 2023-07-20
**Review for FETV paper**

**Rating:** 5
**Confidence:** 4
**Correctness:** This work provides a corresponding Gi…
**Clarity:** well written

**Strengths:**

1. Collected 619 videos and corresponding prompts from the MSR-VTT test set and WebVid, and provided dataset distribution statistics, including prompt complexity distribution statistics and major content distribution statistics.

2. The work includes a substantial amount of experimental analysis, such as manual evaluation of static and temporal video quality (Figure 5), and metric evaluation (Table 3). It also provides some insights, stating that although the automatic metrics are positively correlated with manual evaluations, the overall correlation degree is weak, being lower than 50% of inter-human correlations.

**Additional Feedback:**

None

**Documentation:**

This work provides a corresponding GitHub repository but lacks some maintenance plans.

**Limitations:**

The paper has discussed corresponding limitations

**Opportunities For Improvement:**

1. Unique. The overall impression I get from this benchmark is that it has some uniqueness, but it is not entirely well-explained. Compared to previous benchmarks like Make-a-Video-Eval and MSR-VTT, the main uniqueness of this work lies in its emphasis on evaluating temporal information. However, in this aspect, the authors seem to have not provided strong new insights.  Regarding human evaluation, the assessment involves simple manual grading scores, which may not provide significant inspiration for future works and appears more like a secondary aspect of the algorithmic work. As for metric evaluation, there is insufficient analysis, especially concerning the CLIPScore trained via video-text pair comparison. This has the potential to be a highlight of the work in evaluating video generation tasks, but it is not adequately elaborated.

2. Contribution. The contribution of this paper appears to be relatively weak. Firstly, the human evaluation is not a particularly analyzable perspective, as it is challenging to generalize due to the variation in evaluators invited by different organizations. Additionally, the cost of creating the dataset and conducting human evaluation is not high. Compared to manual evaluation, the reviewers are more interested in CLIPScore and BLIPScore because the CLIPScore trained via video-text pair may potentially offer better video generation evaluation results and possess a stronger perception of temporal information compared to the original CLIPScore. However, the authors have not provided new insights or corresponding analyses in this regard.

In summary, the current version of this work does not present significant contributions, and the evaluation metrics do not seem to attract future video generation models to use them as standard evaluation metrics.



**Relation To Prior Work:**


It seems that some related works are missing:


[1] Wu, Jay Zhangjie, Yixiao Ge, Xintao Wang, Weixian Lei, Yuchao Gu, Wynne Hsu, Ying Shan, Xiaohu Qie, and Mike Zheng Shou. "Tune-a-video: One-shot tuning of image diffusion models for text-to-video generation." arXiv preprint arXiv:2212.11565 (2022).

[2] Molad, Eyal, Eliahu Horwitz, Dani Valevski, Alex Rav Acha, Yossi Matias, Yael Pritch, Yaniv Leviathan, and Yedid Hoshen. "Dreamix: Video diffusion models are general video editors." arXiv preprint arXiv:2302.01329 (2023).

[3]Yang, Ruihan, Prakhar Srivastava, and Stephan Mandt. "Diffusion probabilistic modeling for video generation." arXiv preprint arXiv:2203.09481 (2022).

[4] Blattmann, Andreas, Robin Rombach, Huan Ling, Tim Dockhorn, Seung Wook Kim, Sanja Fidler, and Karsten Kreis. "Align your latents: High-resolution video synthesis with latent diffusion models." In Proceedings of the IEEE/CVF Conference on Computer Vision and Pattern Recognition, pp. 22563-22575. 2023.

**Summary And Contributions:**

This work proposes a video edit dataset, including 619 prompts and corresponding videos. Additionally, it provides a substantial amount of experimental analysis, including both human evaluation and metric evaluation (CLIPScore).

---

> ### Author Response · Authors · 2023-08-26
>
> ## 1. Difference from existing datasets
> ### Reviewer Comment
>     The overall impression I get from this benchmark is that it has some uniqueness, but it is not entirely well-explained. Compared to previous benchmarks like Make-a-Video-Eval and MSR-VTT, the main uniqueness of this work lies in its emphasis on evaluating temporal information. However, in this aspect, the authors seem to have not provided strong new insights.
> ### Response
> Thanks for highlighting this concern. We would like to clarify that our proposed FETV is different from Make-a-Video-Eval and MSR-VTT in **two perspectives**.
>
> Firstly, as you have mentioned, FETV is **temporal-aware**. With the proposed temporal categories, we are able to evaluate (1) the video quality when generating different types of temporal contents and (2) the ability of T2V models to control different types of temporal attributes. In this way, we identify the temporal categories that are challenging for existing T2V generation models, which can serve as useful guidance for T2V model development in the future. We would like to humbly emphasize that this contribution is especially important to the T2V generation and is neglected by Make-a-Video-Eval and MSR-VTT.
>
> Secondly, FETV is **multi-aspect**. The ability to control detailed attributes and handle prompts with different complexity levels is critical for T2V generation models. However, Make-a-Video-Eval and MSR-VTT only categorize prompts based on major contents, lacking the ability to evaluate these two aspects. In comparison, FETV, with its multi-aspect categorization, can effectively assess them.
>
> With the above two advantages, FETV is **more comprehensive than other related benchmarks** (Reviewer 5H2c) and **gives a holistic view of T2V models' capabilities** (Reviewer gcDZ).
>
> ## 2. Simple manual grading scores may not provide significant inspiration
> ### Reviewer Comment:
>     Regarding human evaluation, the assessment involves simple manual grading scores, which may not provide significant inspiration for future works and appears more like a secondary aspect of the algorithmic work.
> ### Response
> We respectfully disagree. In fact, the Likert-scale human grading scores are sufficient to reflect the fine-grained performance across different categories under the proposed three aspects. By splitting the FETV dataset into different subsets based on our categorization, the result of each subset can reflect the performance of the corresponding category. In this way, we identify the categories that are challenging for existing T2V generation models, which we believe is a valuable inspiration for future studies.
>
> ## 3. Insufficient analysis of automatic metrics
> ### Reviewer Comment
>     As for metric evaluation, there is insufficient analysis, especially concerning the CLIPScore trained via video-text pair comparison. This has the potential to be a highlight of the work in evaluating video generation tasks, but it is not adequately elaborated.
> ### Response
> To enhance the reliability of automatic metrics, we employed an advanced video-language model, namely UMT [1], to evaluate the video-text alignment. UMT is pre-trained with video-text data, which is more suitable to T2V generation evaluation than CLIP and BLIP. We find that the UMT-based metric is consistent with the human ranking of T2V models. Please refer to the general response for the detailed results and analysis.

---

> > ### Author Response · Authors · 2023-08-26
> >
> > ## 4. Human evaluation is hard to generalize
> > ### Reviewer Comment
> >     The human evaluation is not a particularly analyzable perspective, as it is challenging to generalize due to the variation in evaluators invited by different organizations.
> > ### Response:
> > We can fully understand this concern, and we would like to assure you that the key findings obtained from our human evaluation are generalizable, because:
> >
> > - We have carefully designed the human evaluation protocol, devising concrete definitions for each rating level and providing some rating examples as reference (Fig.3 in the Appendix), which can reduce the subjective factors of each evaluator. Research on human evaluation of text-to-image generation [2] has also demonstrated that more specific questions and definitions of rating levels can lead to significantly better inter-annotator agreement (IAA).
> > - Our human evaluators are well-qualified. They are experienced in related research fields (T2VI generation and vision-language understanding) and have good English skills.
> > - Our inter-annotator agreement is strong. The IAA in static quality, temporal quality and video-text alignment are 0.711, 0.770 and 0.638, respectively.
> >
> > Additionally, we would like to note that it is necessary to use manual evaluation to ensure an accurate assessment of the T2V generation models and obtain reliable conclusions. This is because existing automatic evaluation metrics agree poorly with human judgement of video/image quality and the alignment with texts, as demonstrated by our experiments and previous findings [2,3].
> >
> > ## 5. Missing Related Work
> > ### Reviewer Comment
> >     It seems that some related works are missing
> > ### Response
> > Thank you for the pointers to these interesting papers. We will appropriately acknowledge and reference these works in our forthcoming revision of the paper.
> >
> > [1] Unmasked Teacher: Towards Training-Efficient Video Foundation Models. ICCV23.
> >
> > [2] Toward Verifiable and Reproducible Human Evaluation for Text-to-Image Generation. CVPR23.
> >
> > [3] On aliased resizing and surprising subtleties in GAN evaluation. CVPR22.

---

> > ### Author Response · Authors · 2023-08-29
> > **Further Response to "1. Difference from existing datasets"**
> >
> > We would like to further discuss a difference from the category distribution perspective to illustrate the advantage of FETV over MSR-VTT and WebVid as a benchmark for T2V generation. To obtain the distribution of MSR-VTT and WebVid, we randomly sample 100 data from each dataset and manually annotate the categories.
> >
> >   As we can see in the following results, FETV exhibits a more uniform distribution over the categories, while MSR-VTT and WebVid are more biased toward certain categories. Particularly, more than 60\% of the MSR-VTT and WebVid prompts do not involve the six attributes introduced in FETV. Consequently, the ability to control these categories, especially the challenging ones, cannot be reflected by MSR-VTT and WebVid (note that these two datasets lack categorization of the attributes).
> >
> > These results and discussions are also provided in the newly updated paper appendix.
> >
> > ### Distribution over attributes to control
> >
> > |  | Color | Camera View | Quantity | Speed | Motion Direction | Event Order | None |
> > | :---: | :---: | :---: | :---: | :---: | :---: | :---: | :---: |
> > | MSR-VTT+WebVid | 15.5% | 10% | 5% | 6% | 9% | 1% | 63% |
> > | FETV | 22.3% | 14.9% | 10.3% | 18.3% | 15.3% | 12.0% | 26.8% |
> >
> > ### Distribution over spatial major contents
> >
> > |  | People | Animals | Vehicles | Artifacts | Plants | Buildings | Food | Scenery | Illustrations | None |
> > | :---: | :---: | :---: | :---: | :---: | :---: | :---: | :---: | :---: | :---: | :---: |
> > | MSR-VTT+WebVid | 61% | 5.5% | 12.5% | 33% | 8% | 19.5% | 5% | 22% | 1.5% | 4.5% |
> > | FETV | 47.7% | 17% | 21.2% | 25.5% | 8.6% | 14.9% | 12.4% | 27% | 3.1% | 0% |
> >
> > ### Distribution over temporal major contents
> >
> > |  | Actions | Kinetic Motoins | Fluid Motions | Light Change | None |
> > | :---: | :---: | :---: | :---: | :---: | :---: |
> > | MSR-VTT+WebVid | 64.5% | 23% | 14.5% | 6.5% | 12.5% |
> > | FETV | 61.4% | 49.3% | 34.7% | 15.5% | 0% |

---

### Official Review · Reviewer_gcDZ · 2023-07-21
**This paper presents a benchmark for fine-grained evaluation of T2V models. While it introduces a new dataset, the contribution is limited due to the absence of novel evaluation metrics.**

**Rating:** 6
**Confidence:** 4

**Strengths:**

The paper introduces a new benchmark, thoughtfully crafted to evaluate open-domain text-to-video generation models. Notably, it goes beyond previous benchmarks by incorporating the fine-grained temporal aspect of video generation, giving a holistic view of T2V models' capabilities. The experiments conducted are rigorous, and the analysis provided offers valuable insights, contributing to the advancement of T2V research.


**Additional Feedback:**

1. The term "unusual prompt" is not rigorously defined in the paper.
2. The problem of “lack of reliable automatic evaluation metrics” remains unsolved.

**Clarity:**

The presentation is clear.


**Correctness:**

The claims made in the submission are correct.


**Documentation:**

The documentation is well written.


**Ethics:**

No.

**Limitations:**

This paper reveals the mismatch between existing automatic metrics and human preferences in the evaluation of T2V models. However, it still relies on manual evaluation. As a benchmark paper, it is important to devise and implement a suitable automatic metric, thereby alleviating the dependency on human annotation and making the evaluation process more efficient and cost-effective.



**Opportunities For Improvement:**

To elevate the quality of this paper, the authors should take into account benchmarking the most recent T2V models, such as Gen-1 and Zeroscope. This addition will provide valuable insights into the performance of cutting-edge models. Moreover, incorporating automatic metrics is imperative to simplify the evaluation process and make it more efficient, reducing the reliance on manual assessments. By implementing these improvements, the paper will become even more robust and valuable to the T2V research community.

**Relation To Prior Work:**

The difference from previous works is clearly discussed.


**Summary And Contributions:**

This paper introduces the FETV benchmark, designed to evaluate open-domain text-to-video generation with a focus on fine-grained analysis. Unlike previous benchmarks, FETV is unique in its multi-aspect and temporal-aware nature, allowing for more detailed insights into the performance of T2V models. By leveraging FETV, the authors conducted a thorough manual evaluation of various T2V models, uncovering their strengths and weaknesses across different categories of text prompts from various perspectives. Additionally, FETV was extended to assess the reliability of automatic T2V metrics, revealing a weak correlation with human evaluations.

---

> ### Author Response · Authors · 2023-08-26
>
> ## 1. Benchmarking the most recent T2V models (Gen-1 and Zeroscope)
> ### Reviewer Comment
>     To elevate the quality of this paper, the authors should take into account benchmarking the most recent T2V models, such as Gen-1 and Zeroscope. This addition will provide valuable insights into the performance of cutting-edge models.
> ### Response
> Thanks for the constructive feedback! In response to the request, we are evaluating a more recent model ZeroScope [1] and the results will be reported in the next version of our paper. However, we would also like to note that ZeroScope is released on Jun 3 after the abstract submission deadline (Jun 1) and Gen-1 does not support text-to-video generation. At the time of writing this paper, we only have access to CogVideo, Text2Video-Zero and VideoFusion.
>
> ## 2. Incorporating automatic metrics
> ### Reviewer Comment
>     Moreover, incorporating automatic metrics is imperative to simplify the evaluation process and make it more efficient, reducing the reliance on manual assessments. By implementing these improvements, the paper will become even more robust and valuable to the T2V research community.
> ### Response
> We fully acknowledge the significance of reliable automatic metrics in the evaluation and development of T2V generation models. To facilitate the usage of FETV for follow-up studies, we employed an advanced video-language model, namely UMT [2], to evaluate the video-text alignment. We find that the UMT-based metric is consistent with the human ranking of T2V models. Please refer to the general response for details.
>
> ## 3. The term "unusual prompt" is not rigorously defined
> ### Reviewer Comment
>     The term "unusual prompt" is not rigorously defined in the paper.
> ### Response
> Our definition of “unusual prompts” is inspired by the “Imagination” category in PartiPrompts [3] and the “Conflicting” category in DrawBench [4], which describe participants or interactions that are not, or are generally unlikely to be, found in the modern day world [3]. Specifically, we consider unusual interactions of “object-action”, “object-object”, “object-color”, “object-quantity”, “object-direction”, “object-speed” and “event order” as described in the paper. We will add citations to [3,4] when introducing the unusual prompts in the next version of the paper.
>
> [1] https://huggingface.co/cerspense/zeroscope_v2_576w
>
> [2] Unmasked Teacher: Towards Training-Efficient Video Foundation Models. ICCV23.
>
> [3] Scaling autoregressive models for content-rich text-to-image generation.
>
> [4] Photorealistic text-to-image diffusion models with deep language understanding.

---

### Official Review · Reviewer_HZhg · 2023-07-21
**Review for A Benchmark for Fine-Grained Evaluation of Text-to-Video Generation**

**Rating:** 6
**Confidence:** 3
**Clarity:** Yes. It’ easy to understand the paper.

**Strengths:**

1. The work proposes a novel benchmark for fine-grained evaluation of T2V generation in a comprehensive view.

2. Their research reveals that the existing metrics correlate poorly with human judgments, giving other researchers evidence to improve T2V generation.

**Additional Feedback:**

1. The reference of WebVid should be cited, and the references of the three models CogVideo, Text2Video-zero and VideoFusion should also be cited separately.

2. It’s better to invite some other people to evaluate the videos, not only just students.

3. It’s better to state the related works earlier so that readers can have better preliminary knowledge of T2V evaluation.


**Correctness:**

Correct. It’s better to invite some other people to evaluate the videos, not only just students.

**Documentation:**

Yes.

**Ethics:**

No ethical concerns.

**Limitations:**

Although they reveal the reliability problem of automatic T2V evaluation metrics and show some potential solutions for improvement, they do not present a specific automatic metric that can reliably reflect human judgments.

**Opportunities For Improvement:**

1. Categories in the three aspects can be richer and more detailed.

2. Despite the contributions, it is better to propose a specific solution such as a specific automatic metric that can reliably reflect human judgments.

**Relation To Prior Work:**

Yes, compared with other works, FETV is both multi-aspect and temporal-aware, which can provide more specific information on the performance of T2V models.

**Summary And Contributions:**

The paper proposes the FETV benchmark for fine-grained evaluation of open-domain text-to-video generation, which includes three essential parts: major content, attribute control and prompt complexity. They use the MSR-VTT test set and WebVid validation set to collect data, do manual evaluations on three open-source T2V models, CogVideo, Text2Video-zero and VideoFusion, and reveal their pros and cons from different aspects. In addition, they extend FETV as a testbed to evaluate the reliability of automatic T2V metrics and reveal that the existing metrics correlate poorly with human judgments.

---

> ### Author Response · Authors · 2023-08-26
>
> ## 1. Categories can be richer and more detailed
> ### Reviewer Comment
>     Categories in the three aspects can be richer and more detailed.
> ### Response
> Thanks for the suggestion! Indeed, under the multi-aspect structure, we can extend FETV to cover a wider range of categories. However, we believe that our current categorization is already comprehensive and challenging enough to evaluate existing T2V generation models.
>
> The “major content” categories of FETV consist of the most common categories based on our observation of the MSR-VTT and WebVid video captions. We randomly sample 200 captions from MSR-VTT and WebVid, and investigate whether they involve “unseen spatial or temporal contents” that cannot be classified into one of our defined categories. We find that there are only 11 such captions involving unseen spatial contents and none of the 200 captions involve unseen temporal contents. The 200 sampled captions with annotation of unseen spatial and temporal contents can be found in this link (https://github.com/llyx97/FETV/blob/main/sampled_WebVid_MSRVTT.json).
>
> While the current “attribute control” categories in FETV focus on basic attributes to control, they already pose great challenges to existing T2V generation models, as shown by the results of Section 3.3. We leave the evaluation of more difficult attributes, e.g., spatio-temporal relationship between objects, to future work.
>
> ## 2. Specific automatic metric that can reliably reflect human judgments
> ### Reviewer Comment
>     Despite the contributions, it is better to propose a specific solution such as a specific automatic metric that can reliably reflect human judgments.
> ### Response
> We totally agree on this point and we have employed a more advanced video-language model, namely UMT [1], to evaluate the video-text alignment. We find that the UMT-based metric is consistent with the human ranking of T2V models. Please refer to the general response for details.
>
> ## 3. It's better to invite other people for manual evaluations
> ### Reviewer Comment
>     It’s better to invite some other people to evaluate the videos, not only just students.
> ### Response
> We agree that inviting more evaluators can facilitate the reliability of the results. However, we believe that the key findings obtained from our human evaluation are already reliable, because:
>
> - We have carefully designed the human evaluation protocol, devising concrete definitions for each rating level and providing some rating examples as reference (Fig.3 in the Appendix), which can reduce the subjective factors of each evaluator. Research on human evaluation of text-to-image generation [2] has also demonstrated that more specific questions and definitions of rating levels can lead to significantly better inter-annotator agreement (IAA).
> - Our human evaluators are well-qualified. They are experienced in related research fields (T2VI generation and vision-language understanding) and have good English skills.
> - Our inter-annotator agreement is strong. The IAA in static quality, temporal quality and video-text alignment are 0.711, 0.770 and 0.638, respectively.
>
> ## 4. Citation of WebVid, and three T2V models
> ### Reviewer Comment
>     The reference of WebVid should be cited, and the references of the three models CogVideo, Text2Video-zero and VideoFusion should also be cited separately.
> ### Response
> Thanks for the feedback. We will properly cite these works in our forthcoming revision of the paper.
>
> ## 5. State related work earlier
> ### Reviewer Comment
>     It’s better to state the related works earlier so that readers can have better preliminary knowledge of T2V evaluation.
> ### Response
> Thank you for the suggestion. We will move the related work section before introducing the FETV benchmark in the next version.
>
> [1] Unmasked Teacher: Towards Training-Efficient Video Foundation Models. ICCV23.
>
> [2] Toward Verifiable and Reproducible Human Evaluation for Text-to-Image Generation. CVPR23.

---

### Author Response · Authors · 2023-08-26
**General Response**

Dear reviewers,

We thank you so much for the detailed and constructive feedback! It's encouraging to note the contributions of our work highlighted by reviewers: the novelty and comprehensiveness of the proposed benchmark (Reviewer HZhg, gcDZ and 5H2c) and valuable insights from the experiments (Reviewer HZhg, gcDZ, 5H2c and 2CxJ). We have thoroughly read the review comments and updated experiments and discussions accordingly, which will be incorporated in the next version of the paper.

Here we address some common questions raised by the reviewers:

## 1. Lack of reliable automatic evaluation metrics
We agree that it is crucial to have reliable automatic metrics for a benchmark. To this end, we employed an advanced video-language model named UMT [1], to evaluate the video-text alignment. UMT is pre-trained on large-scale video-text data and we fine-tune it on the MSRVTT dataset using Video-Text Contrastive (VTC) learning and Video-Text Matching (VTM) losses. To obtain the alignment score between a pair of text and video, we use the VTM result from the output of cross-modal decoder of UMT and name this metric as the "UMT-MatchScore". As shown in the following results, the correlation between UMT-MatchScore and human judgements is significantly higher than BLIPScore and CLIPScore-ft. More importantly, when it comes to the ranking of T2V generation models, UMT-MatchScore is the only metric that is consistent with humans. Therefore, we believe that UMT-MatchScore, while still having room for improvement, can be used as a reliable metric to evaluate existing T2V generation models. Details of the correlation results in each category and our implementation of UMT-MatchScore will be provided in the next version of our paper.

### Correlation with human evaluation.
|Metrics  | Kendall | Spearman |
| --- | --- | --- |
| CLIPScore-ft | 23.3 | 32.3 |
| BLIPScore | 24.8 | 34.0 |
| UMT-MatchScore | **32.9** | **45.1** |
| Inter-Human | 58.0 | 73.0 |
### Ranking of T2V generation models by different metrics
|  | Real | VideoFusion | T2V-Z | CogVideo |
| --- | :---: | :---: | :---: | :---: |
| CLIP | 0.306(2) | 0.315(1) | 0.304(3) | 0.285(4) |
| CLIP-FT | 0.299(2) | 0.298(3) | 0.308(1) | 0.260(4) |
| BLIPScore | 0.447(3) | 0.471(1) | 0.462(2) | 0.411(4) |
| UMT-MatchScore | 2.780(1) | 2.496(2) | 2.181(3) | 1.023(4) |
| Human | 4.887(1) | 3.787(2) | 3.416(3) | 3.103(4) |

## 2. Evaluation of more T2V generation models
We fully understand that the reviewers expect evaluation of more T2V models. However, we would like to note that the three models we evaluated are the only ones that are open-sourced at the writing of the paper. Zeroscope_v1 was released in Hugging Face on June 3, 2023 [2] and Gen-2 was available on Jun 7, 2023 [3], both later than the abstract submission deadline (Jun 1, 2023). Gen-1 was released earlier, but it does not support text-to-video generation [4].

To better understand the ability of the most recent T2V generation models, we are evaluating Zeroscope_v2_576w [5] on our FETV benchmark and the results will be reported in the revised paper.

[1] Unmasked Teacher: Towards Training-Efficient Video Foundation Models. ICCV23.

[2] https://huggingface.co/cerspense/zeroscope_v1_320s

[3] https://twitter.com/runwayml/status/1666429706932043776

[4] https://research.runwayml.com/gen1

[5] https://huggingface.co/cerspense/zeroscope_v2_576w

---

> ### Author Response · Authors · 2023-08-31
> **Introduction of a Reliable Automatic Video Quality Metric**
>
> Hi reviewers,
>
> We present a new automatic video quality metric based on the UMT model [1]. This metric, which we call FVD-UMT, modifies the FVD metric by replacing the original I3D video feature extractor [2] with the UMT vision encoder. As shown in the following results, FVD-UMT is consistent with the human ranking of the three T2V generation models in terms of video quality (computed as the average of SQ and TQ). More details can be found in the newly uploaded paper and appendix.
>
> With FVD-UMT and the UMTScore we presented in the first General Response, we can automatically and reliably evaluate the T2V models on FETV, in terms of both video quality and video-text alignment.
>
>
> |  | VideoFusion | CogVideo | Text2Video-zero |
> | :---: | :---: | :---: | :---: |
> | FID ($\downarrow$)                | 68.9(2) | 61.6(1) | 71.4(3) |
> | FVD ($\downarrow$)                | 673.9(3) | 403.4(1) | 499.3(2) |
> | FVD-UMT ($\downarrow$)            | 178.7(1) | 210.9(2) | 213.4(3) |
> | Static Quality (SQ) ($\uparrow$)  | 3.78 | 3.24 | 4.11 |
> | Temporal Quality (TQ) ($\uparrow$) | 4.07 | 3.30 | 2.04 |
> | (SQ+TQ)/2  ($\uparrow$)           | 3.93(1) | 3.27(2) | 3.08(3) |
>
> [1] Unmasked Teacher: Towards Training-Efficient Video Foundation Models. ICCV23.
>
> [2] Quo vadis, action recognition? A new model and the kinetics dataset. CVPR17.

---

### Author Response · Authors · 2023-08-29
**Further General Response**

Dear reviewers,

Thank you again for your reviews! We have submitted a revised version of our paper and appendix, with the following updates:
- Results of using the UMTScore as an automatic evaluation metric. (Reviewer HZhg, gcDZ, 2CxJ, 5H2c)
- Human evaluation results of a more recent T2V generation model, ZeroScope. (Reviewer gcDZ and 5H2c)
- Provide FETV data examples, generated videos and human ratings in our dataset repo (https://github.com/llyx97/FETV/tree/main/generated_video_examples), which may give the readers a more intuitive perception of our data quality and the pros and cons of existing T2V models (Reviewer 5H2c).
- Compare FETV with MSR-VTT and WebVid in terms of the distribution over categories, as an additional detail to illustrate our dataset quality. (Reviewer 5H2c, 2CxJ)
- Clarify the quality of our human evaluations. (Reviewer 5H2c)
- Clarify the definition of the "unusual prompts". (Reviewer gcDZ)
- Incorporate some missing related work. (Reviewer 2CxJ)
- Place the related work section before the FETV benchmark section. (Reviewer HZhg)
- Citation of WebVid and separate citation of CogVideo, Text2Video-zero and VideoFusion. (Reviewer HZhg)

The updated contents are marked in blue on paper to facilitate reading.

---

### Decision · Program_Chairs · 2023-09-22

**Decision:**

Accept (Poster)

**Comment:**

Three of the four reviewers support borderline accept (score 6). One reviewer 2CxJ scores marginally below acceptance (5). The main concerns of 2CxJ are (1) the uniqueness of the dataset and (2) weak contributions especially on manual grading score vs. CLIPScore and BLIPScore, and on lacking an automatic metric (also pointed out by other reviewers). The authors have responded extensively. Unfortunately reviewer 2Cx didn't follow-up on the authors comprehensive responses to raise any remaining issues. I consider that the two issues of existing text-to-video datasets the authors identified are important and this dataset is timely. It does offer two critical unique perspectives in both temporal-aware and multi-aspect as compared to existing datasets. Moreover, the authors' explanation on the value of manual grading score is persuasive, and the added automatic metric for evaluation in the revised paper is credible. On balance I recommend accept.